# Do verbal coaching cues and analogies affect motor skill performance in youth populations?

Jason Moran[1]*, Raouf Hammami[2,3], Joshua Butson[1], Matt Allen[4], Abdelkader Mahmoudi[2], Norodin Vali[5], Ieuan Lewis[6], Phil Samuel[6], Mike Davies[7], James Earle[8], Megan Klabunde[9], Gavin Sandercock[1]

1 School of Sport, Rehabilitation, and Exercise Sciences, University of Essex, Colchester, United Kingdom, 2 Higher Institute of Sport and Physical Education of Ksar-Said, Universite de La Manouba, Tunis, Tunisia, 3 Research Laboratory: Education, Motor Skills, Sports and Health, Higher Institute of Sport and Physical Education of Sfax, University of Sfax, Sfax, Tunisia, 4 Tottenham Hotspur Football Club, London, United Kingdom, 5 Department of Exercise Physiology, Shahid Rajaee Teacher Training University, Tehran, Iran, 6 Stevenage Football Club, Stevenage, Hertfordshire, United Kingdom, 7 Abingdon School, Oxfordshire, United Kingdom, 8 Culford School, Suffolk, United Kingdom, 9 Department of Psychology, University of Essex, Colchester, United Kingdom

* jmorana@essex.ac.uk

**Data Availability Statement:** All relevant data are within the paper and its Supporting Information files

# Abstract

## Purpose

The way coaching cues are worded can impact on the quality with which a subsequent motor skill is executed. However, there have been few investigations on the effect of coaching cues on basic motor skill performance in youths.

## Method

Across several international locations, a series of experiments were undertaken to determine the effect of external coaching cues (EC), internal coaching cues (IC), analogies with a directional component (ADC) and neutral control cues on sprint time (20 m) and vertical jump height in youth performers. These data were combined using internal meta-analytical techniques to pool results across each test location. This approach was amalgamated with a repeated-measures analysis to determine if there were any differences between the ECs, ICs and ADCs within the different experiments.

## Results

173 participants took part. There were no differences between the neutral control and experimental cues in any of the internal meta-analyses except where the control was superior to the IC for vertical jump (d = -0.30, [-0.54, -0.05], p = 0.02). Just three of eleven repeated-measures analyses showed significant differences between the cues at each experimental location. Where significant differences were noted, the control cue was most effective with some limited evidence supporting the use of ADCs (d = 0.32 to 0.62).

**Funding:** The funders of our study which was the University of Essex which awarded faculty funds to Dr Jason Moran for this project in the amount of £5,980. Moran, Sandercock, Butson and Klabunde are all employed in full time research roles by the funders, the University of Essex. Therefore employees of the funders undertook the study design, data collection and analysis, decision to publish and preparation of the manuscript. In this way, the study was internally funded by the employers of several of the research team.

**Competing interests:** The authors have declared that no competing interests exist.

## Conclusion

These results suggest the type of cue or analogy provided to a youth performer has little subsequent effect on sprint or jump performance. Accordingly, coaches might take a more specific approach that is suited to the level or preferences of a particular individual.

## Introduction

A coaching cue is a verbal instruction that can be used to focus an individual's attention on a movement to optimise its execution [1]. Cues that direct a performer's attention externally (focus placed outside of the body) or internally (focus on body part) have been shown to have effects on subsequent motor skill performance [2–4]. The constrained action hypothesis [5] suggests that an external focus of attention can result in improved motor performance by increasing the automaticity of movement control during an action [6]. It has been proposed that an external focus reduces the attentional capacity that is needed to carry out a movement which can also be enhanced through greater coordination between working muscles [7]. Accordingly, the manner in which coaching cues are worded and presented to performers can immediately impact on the quality with which motor skills are executed [8] and in the longer term, this can be reinforced through the learning process [9].

Expanding on the above concept, and citing the work of Porter [2, 10], Winkelman [11] highlighted the impact of using a directional component to enhance motor performance. It is explained that cues that include a distal focus of attention appear to be more effective in driving jumping performance than those with a proximal focus. Winkelman recontextualised a proximal focus as an "away-focus" (i.e. "jumping as far past the start line as possible") and a distal focus as a "toward-focus" (i.e. "jumping as close to the cone as possible") [11]. In this way, it is suggested that an individual might demonstrate better performance when presented with a distal, or "toward", focus that fixes their attention on a point or target that is external to their body, or is present in the environment around them [11]. It is generally accepted that the use of external coaching cues (EC) can lead to positive performance outcomes in adults with an expectation that similar results would be seen in youth performers [12]. Accordingly, the utilisation of the above explained concepts as a tools for the coaching of skills such as running and jumping, may represent a logical next step in advancing practice in this domain.

Despite the above, to date, there have been very few investigations on the effect of various different foci of attention on the performance of skills such as running and jumping, in youths. This has made it particularly difficult to determine whether or not the manipulation of attentional focus through coaching cues can enhance performance and subsequent motor learning in young individuals. Accordingly, coaches therefore might not be using optimal methods when attempting to drive motor learning in these populations. Two systematic reviews [4, 13] *have* provided a comprehensive overview of the various studies that have been undertaken in this area, however, of the investigations carried out in children, most related to sport-specific skills such as basketball dribbling, golf putting and tennis serving, as opposed to basic skills such as jumping. Moreover, not a single study in children has examined the effect of different coaching cues on the execution of running or sprinting, vital skills as youths develop. This is particularly concerning given that physical literacy has been associated with higher levels of physical activity [14] meaning that if youths can achieve movement proficiency, they may be more likely to engage in physical activity across the lifespan [15].

A largely unexplored advancement on the concept of using an external focus of attention to enhance motor performance and learning is the use of analogies to convey the objective of a

given coaching cue. An analogy is a coaching instruction that conceals biomechanical cues within spoken words. It differs from a conventional instruction in that it conveys key elements of a given movement without the need to specifically reference those same elements [16]. In sport, analogies can be used to demonstrate the required body position and speed for a given action, representing movement in a symbolic way that is potentially more understandable to an athlete [17]. Recently, Fasold et al. [18] found that children exhibited improved performance in handball skills when coaching cues were delivered in an analogical format. Indeed, it has been suggested that this method of coaching delivery should be prioritised when working with young athletes on the basis that it can improve information processing by enhancing the recall of instructions, thus making those instructions more relatable to the task to which they specifically refer [19, 20].

The above described concepts could be useful in the context of coaching and teaching. However, to date, research that investigates the effects of a directional component (i.e. 'towards' vs. 'away') is sparse [21]. Moreover, to the authors' knowledge, no previous study has examined these types of foci when combined with analogies in any population, let alone in youths; this despite the merging of ECs with analogies being suggested to be an effective way to retain focus for optimal performance [6, 8]. This might have important implications for coaching and learning as the combination of ECs and analogies could represent a previously known, yet untested, tool for coaches that could enhance the contextual understanding of a performer and, by extension, the learning of key movement skills.

The purpose of this research was to determine the effectiveness of internal cues (IC), ECs and external 'analogies with a directional component' (ADC) on motor skill performance in youths in various different populations, ranging from school children to academy athletes, and across a variety of international contexts and languages. On the basis of the above described literature, it was hypothesised that ECs, ICs and ADCs would be more effective than neutral control cues, that ECs would be more effective than ICs and that ADCs would be more effective than both ECs and ICs, at enhancing vertical jump and 20 m sprint.

## Methods

### Experimental design

Participants undertook ten vertical jumps and 20 m sprints prior to which they were given a specific coaching cue relating to their performance. Several similar, yet separate, experiments were conducted on the effect of ECs, ICs and ADCs on motor performance in youths across a variety of domestic (UK) and international centres. 173 participants were recruited from a variety of different backgrounds, developmental levels and ages. The descriptive characteristics of the various groups can be viewed in Table 1. Only individuals under the age of 18 were eligible to take part and though the study was open to female participants, it was not possible to recruit any into the various cohorts. Only healthy individuals were considered and the study was open to both trained and untrained participants. The various experiments were carried out across diverse a group of youths such as 14-year old English-speaking grammar school students in the UK, 10-year old French-speaking academy soccer players in Tunisia and 15 year old Persian-speaking soccer players in Iran. The coaching cues that were delivered in French and Persian are in the supplemental information to this study.

We utilised an internal meta-analytical approach to pool the gathered data from across the various centres. This method of meta-analysing one's own studies has previously been advocated on the basis that it provides a clearer consensus on a given topic and enhances statistical power because individual studies can be underpowered when evaluated in isolation [22]. With so few studies having been undertaken on the effect of verbal coaching cues for performance

**Table 1. Descriptive characteristics of the study groups.**

| Population, age, language | Sex | Training experience | Surface used | Sprint apparatus | Jump apparatus |
|---|---|---|---|---|---|
| Soccer academy players (10 yrs, French) | Male | 2–3 years | Grass | TCI System, Brower Timing Systems, Utah, United States | OptoJump Next, Microgate, Bolzano, Italy |
| Boarding school students (14.5 yrs, English) | Male | 1–4 years | Grass | Witty timing system, Microgate, Bolzano, Italy | |
| Rugby academy students (18 yrs, English) | Male | 4–5 years | Indoor hard basketball court | TCI System, Brower Timing Systems, Utah, United States | ForceDecks Dual Force Plate System, VALD Performance, Queensland, Australia |
| League 2 academy soccer players (10–11 yrs, English) | Male | | Artificial grass | TCI System, Brower Timing Systems, Utah, United States | Chronojump, Boscosystem, Barcelona Spain |
| Grammar school students (14 yrs, English) | Male | | Indoor hard basketball court | TCI System, Brower Timing Systems, Utah, United States | ForceDecks Dual Force Plate System, VALD Performance, Queensland, Australia |
| Youth soccer players (15 yrs, Persian) | Male | 2–3 years | Artificial grass | Newtest Powertimer, 300-series, Oy, Finland | Sargent Jump Test |

in youths, this approach was justified as it would expedite the acquisition of knowledge in the area. This approach was combined with a within-country repeated measures analysis to determine the effect of the various coaching cues on performance at each of the six centres involved in the study. The design of the research can be viewed in Fig 1 below. The research was approved by the University of Essex and conformed to the Declaration of Helsinki. Parental consent and participant assent was attained to take part. For some of the included data, secondary anonymised datasets were necessarily provided with the permission of the relevant parties in each location.

Each participant performed ten jumps and ten sprints with a single instructional cue provided to them immediately before each action. There were five different cues for the sprints

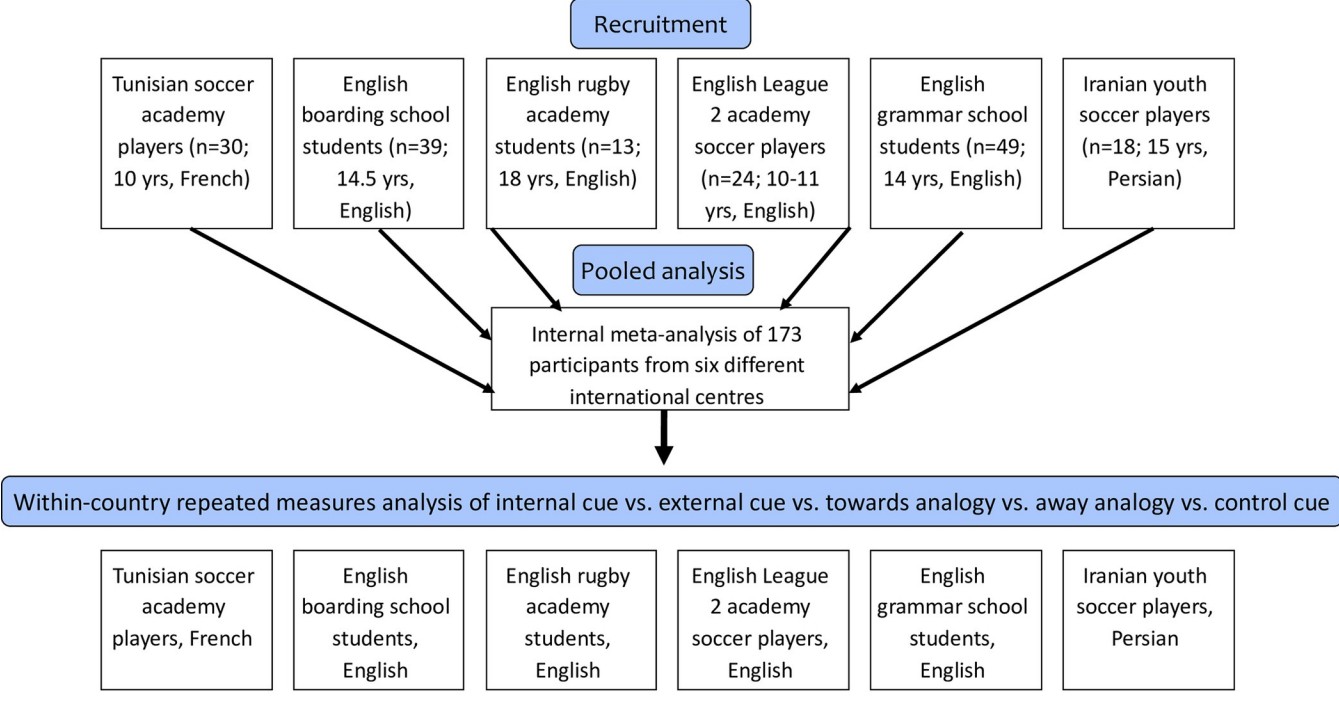

**Fig 1. Research design.**

**Table 2. Jump and sprint cues.**

| Type | Jump cue |
|------|----------|
| Control/neutral | "Jump as high as you can" |
| Internal | "As you jump, focus on extending your legs" |
| External | "As you jump, focus on pushing the ground away" |
| Analogy (away) | "Jump as if the ground is suddenly hot and you have to get off it as quick as possible" |
| Analogy (toward) | "Jump as if you are trying to catch a ball overhead at its highest point" |
| | **Sprint cue** |
| Control/neutral | "Sprint as fast as you can" |
| Internal | "Sprint and focus on driving your legs back" |
| External | "Sprint and focus on driving the ground back" |
| Analogy (away) | "Sprint as if you are being chased up a hill" |
| Analogy (toward) | "Sprint as if you are a jet taking off into the sky ahead " |

and five different cues for the jumps meaning each participant received each individual cue twice. The cues themselves fell into five distinct categories based on type and these can be seen in Table 2 below. They were informed by the work of Winkelman [21]. The terms "jump as high as you can" and "sprint as fast as you can" were neutral cues used as controls against which the various ICs, ECs and ADCs were compared [23, 24].

## Warm up prior to performance

Prior to the fitness tests, a standardised eight minute warm up was carried out following previous protocols [25, 26]. In brief, this included low-intensity running, dynamic movement drills (high knee walks, forward leg swings, overhead lunge walks, straight leg walks, lateral lunges, high knee skips, skip for height) and submaximal jumps and sprints. Prior to performance, participants were permitted to execute two sub-maximal repetitions over five metres and one maximal repetition over ten metres to familiarise themselves with the sprint test format [27]. No cues were provided for these submaximal efforts. Between testing efforts, participants were encouraged to maintain general low intensity movement to remain prepared for performance.

## Jumps

A vertical countermovement jump test was conducted. Prior to any jumps taking place, each participant was individually requested to "jump as high as you can in the remaining ten jumps". They were also informed "prior to each jump you will be given a specific coaching cue. Focus as hard as you can on this cue during the jump" [27]. All cues were read from a seated position that was four feet to the left of the jump position where the participant stood [27]. When jumping, participants executed a downward movement to a self-selected depth/knee flexion angle before performing a vigorous extension of the lower-body limbs to jump as high as possible. The arms were positioned akimbo (i.e. with the hands on the hips and the elbows turned outward) and the feet positioned approximately shoulder width, at a distance comfortable for the participant. There was at least two minutes rest between efforts and each participant's best effort (i.e. highest jump in cm) out of two trials was used in the analysis [28]. When jumping and sprinting were performed on the same day, all jumps were performed first.

## 20-metre sprint

One sprint demonstration was collectively provided for all participants at the start of the session [27]. During this demonstration, the participants were instructed to assume a typical two

point stance, with their feet hip width apart, by placing one foot behind the start line and the other foot back at a comfortable distance. They were requested to position their arms such that they were set opposite from their legs [27]. They were also instructed to "load into your legs and shift forward so that you feel tension and a readiness to sprint forward with no delay" [27]. Prior to any sprints taking place, each participant was individually informed that "the remaining ten sprints will be completed as fast as you can at 100% of your full speed. Prior to each sprint you will be given a specific coaching cue. Focus as hard as you can on this cue during the entire sprint" [27].

All cues were read from a seated position that was four feet to the left of the start line where the participant stood [27]. The test was initiated when the participant voluntarily started the sprint immediately following the provision of one of the instructional cues. At least two minutes of rest was taken between each sprint. The timing gates were set at the start line (0.3 m in front of the subjects), and 20 metres away from the start line. They were positioned 0.7 m above the ground (i.e., hip level), allowing us to capture trunk movement only and to avoid a false trigger from a limb [29].

## Coaching cues

A Latin square design was used to offset order effects due to fatigue or other factors that could impact participants' performance. Each participant was randomly allocated a specific "order scheme" (between 1 and 10) via a random number generator (https://www.random.org/). This 'order scheme' determined the sequence in which each individual received their instructional cues prior to jumping or sprinting. Each letter corresponded to a particular coaching cue, which can be seen in the supplemental information. As participants sprinted and jumped twice for each cue, each cue appeared twice. The order schemes can also be seen in the supplemental information.

## Statistical analyses

Meta-analytical comparisons were carried out in RevMan version 5.3 [30]. Means and standard deviations for the measured jumps and sprints were used to calculate an effect size (standardised mean difference). The neutral control cue was compared separately against the ECs, ICs and two different ADCs resulting in four analyses for jumps and four for sprints (EC vs control, IC vs control, and ADCs (x2) vs. control). The inverse-variance random effects model for meta-analyses was used because it allocates a proportionate weight to trials based on the size of their individual standard errors [31] and facilitates analysis whilst accounting for heterogeneity across cohorts [32]. Though it was not expected that substantial heterogeneity would be present due to the standardised methodological approach that was adopted, there were minor differences between the data collection methods at each of the different centres in the study. Effect sizes are presented alongside 95% confidence intervals (CI). The calculated effect sizes were interpreted using the conventions outlined for standardised mean difference by Hopkins et al [33] (<0.2 = trivial; 0.2–0.6 = small, 0.6–1.2 = moderate, 1.2–2.0 = large, 2.0–4.0 = very large, >4.0 = extremely large).

After the meta-analytical comparisons, a repeated measures analysis was undertaken to determine if there were any differences between the ECs, ICs and two ADCs used in the study. These analyses were carried out using JASP (version 10.2, University of Amsterdam). Data normality was determined with the Shapiro-Wilk test. A repeated measures ANOVA was used to detect any statistically significant ($p < 0.05$) changes in the dependent variables. Again, the calculated effect sizes were interpreted using the conventions outlined for standardised mean difference by Hopkins et al [33], as in the meta-analytical comparison.

**Table 3. Repeated measures analysis for the various coaching cues for both sprints and jumps.**

| Population | Sprints p-value | Jumps p-value | Significant sprint results (effect sizes [d]) | Significant jump results (effect sizes [d]) |
|---|---|---|---|---|
| Soccer academy players (10 yrs, French) | <0.001 | 0.097 | Con > EC (0.3), Away > Con (0.32), Away > IC (0.35), Tow > EC (0.42), Away > EC (0.62) | n/a |
| Boarding school students (14.5 yrs, English) | 0.634 | n/a | n/a | n/a |
| Rugby academy students (18 yrs, English) | 0.369 | 0.149 | n/a | n/a |
| League 2 academy soccer players (10–11 yrs, English) | 0.603 | 0.297 | n/a | n/a |
| Grammar school students (14 yrs, English) | 0.161 | <0.001 | n/a | Con > IC (0.199), Con > EC (0.15), Con > Away (0.167) |
| Youth soccer players (15 yrs, Persian) | 0.552 | <0.001 | n/a | Con > IC (0.65), Con > EC (0.46), Con > Away (0.57) |

## Results

The supplemental information to this study contains the forest plots for the internal meta-analytical results of the analyses where all experimental cues were compared to the neutral control cue. In summary, there were no significant differences between the neutral control cue and experimental cues in seven of the eight of the analyses. Effect sizes, representing the differences between performance under the various cues, ranged from -0.03 (95% confidence interval: -0.24, 0.18) to 0.07 (95% confidence interval: -0.14, 0.28) for the sprint measures and were all classified as 'trivial'. Effect sizes for the jump measures ranged from -0.30 (95% confidence interval: -0.54, -0.05) to -0.15 (95% confidence interval: -0.40, 0.09). Across all analyses, the only effect size that crossed the threshold from 'trivial' to 'small' was that for the IC ("as you jump, focus on extending your legs") when compared to the neutral control cue ("jump as high as you can") in the jump analysis. In that instance, there was a 'small' statistically significant effect size which favoured the neutral control cue (d = -0.30, 95% confidence interval: [-0.54, -0.05], p = 0.02).

Due to the finding that there were very few differences between neutral control cues, ECs, ICs and ADCs, a further repeated measures analysis was undertaken to determine if there were significant differences when all of these variables were compared to one another within each international centre. Table 3 contains the results of this analysis for the various coaching cues for both sprints and jumps. Just three of eleven analyses (six for jumps and five for sprints) showed significant differences between the coaching cues delivered to the participants at each centre. Where significant differences were noted between coaching cues within each of the cohorts, it appeared that the neutral control cue was the most commonly effective, with the 'away analogy' also being effective when delivered for the sprints in French. No individual cohort demonstrated significant differences between cues in both jumps and sprints.

## Discussion

The purpose of this research was to determine the effectiveness of ICs, ECs and two different ADCs ("towards" and "away") on vertical jump and 20 m sprint performance in youths in various different populations, ranging from school children to academy athletes, and across a variety of international contexts and languages. Previous research has demonstrated that coaching language, such as ECs, can have a positive effect on sprint and jump performance; however most evidence relates to adult rather than youth populations [12]. Accordingly, it was hypothesised that ECs, ICs and ADCs would be more effective than neutral control cues for

enhancing jump and sprint performance, that ECs would be more effective than ICs and that ADCs would be more effective than both ECs and ICs.

The results of this investigation imply that across the age spectrum of different groups of youths, there appear to be very few benefits to performance in manipulating directive language according to the various coaching cues outlined in *this* particular study. No evidence was found to support the hypotheses that ECs, ICs and ADCs would be more effective than neutral control cues or that ECs would be more effective than ICs. Some of the observed evidence did suggest that ADCs are more effective than both ECs and ICs. However, this was observed in the French language only and applied to sprinting and not jumping. Where significant differences were seen, instructing an individual to jump as high, or sprint as fast, as possible (i.e. the neutral control cue) was more commonly effective at eliciting improvements in performance.

A coach's ability to communicate instructions effectively has been proposed as a way of driving motor skill development in youths [12]. In this context, the term "effective" refers to a practitioner's ability to use instructions and verbal cues that are understandable to a trainee [12] who is the "target" of such cues. Indeed, the comprehension of verbal cues appears to be a prerequisite to a youth partaking in physical activity [19]. However, as youths develop as they age, their neurocognitive capacities, as well as their ability and willingness to follow instructions, can vary [19]. This means that when working with youths, the challenges of coaching may be different to those encountered when working with adults [19]. The results of the current study are indicative of this when compared to current literature which indicates that coaching techniques such as ECs result in enhanced sprint and jump performance in adults and youths, though evidence in the latter group is relatively scarce [12]. In seven of the eight analyses in the current study, that compared an EC, an IC or an ADC with a neutral (i.e. control) cue, there were no differences in sprint or jump performances observed. In the only analysis that did deviate from this trend, a neutral jump cue ("jump as high as you can") was superior to an internal cue ("focus on extending your legs") with a small effect size observed. On the basis of these results, it appears that the ECs, ICs and ADCs exerted little to no effect on performance, a surprising result given that it is accepted that such manipulation of language and attention has been used to enhance motor performance in adult populations [4, 12].

The constrained action hypothesis [5] implies that an external focus of attention can result in improved motor performance on the basis that it underpins the automaticity of movement control during a given action and supports implicit learning [6]. Conversely, the use of an internal focus of attention is said to promote conscious control of an action thus impeding automatic control and negatively affecting motor performance [34]. On this basis, a focus on the effects instead of the process of a given action might facilitate a type of self-organisation whereby the motor system bypasses the constraints associated with the conscious control of movement [34]. Based on the extant evidence, these concepts seem to hold in adults [35] yet according to the current results at least, the predicted outcomes of constrained action hypothesis could be impeded in certain youth populations in performance tests such as the vertical jump and 20 m sprint. Why this is the case is not entirely clear; however, there could be a practical explanation for this finding that has previously been raised by Maxwell and Masters [36]. These authors demonstrated that when asked to perform a balance task, performers who were provided with an internal focus of attention had switched to an external focus once they executed the task. Comparative research [37] that has been carried out in children and adults is suggestive of a shorter span of attention in the former group. The reason for this could potentially be explained by the rate of cognitive development in children and adolescents whose frontal lobes continue to mature as they grow [37]. Whilst concepts such as the constrained action hypothesis might serve as an effective model for motor performance and learning in adults, an alternative approach could be more appropriate in certain (perhaps naïve) youth

groups due to the aforementioned factors, though evidence to the contrary does exist [38, 39]. In relation to the current study, it is possible that this process was observed at work with the young participants at the testing centres potentially less receptive to the coaching cues provided by the investigators.

Given that there were no differences in performance when ECs and ICs were used, the results of this study could point to the potentially narrower frame of reference that youths possess in comparison to adults which could, in turn, can have a detrimental impact on their understanding of a given coaching cue [12]. It has been argued that because younger people have fewer past life experiences than adults, they could be classified as 'naïve perceivers' [40]. Accordingly, young peoples' life experiences may not have developed to the extent that they can contextualise instructions in the same way that adults do, particularly if those instructions are accompanied by an analogy that lacks context in terms of their understanding of a particular coaching cue. Moreover, whilst adults have been shown to focus on relevant cues only, children focus on both relevant and irrelevant cues and this could potentially impact on the level of attention they devote to a specific instruction [40]. On the contrary, it is important to consider that if a young individual achieves a sufficient volume and quality of training, it is possible that they would no longer be considered to be a naïve perceiver and so may respond more readily to ADCs or ECs when provided by a coach [38].

We did observe limited evidence that ADCs might be an effective way of driving improved performance in sprinting and jumping in young individuals. The use of analogies in coaching youths may serve as a more relatable model of communication that facilitates a better understanding of a coach's cue than the use of traditional biomechanical terminology [19]. In this way, the evocative language of instructing a young athlete to "take off like a rocket" [6] could be preferable if it results in a better contextual understanding than a cue relating to the movement and angles of specific limbs. However, where ADCs could potentially have drawbacks is in relation to the kinetic and kinematic characteristics of a given movement. For example, Winkelman states that if the information verbalised by a coach is not related to the task-relevant characteristics of a given motor skill, a cue may be less effective [11]. Moreover, as an analogy can contain several different pieces of information in a single cue, it could have a negative effect on working memory during performance [41]. Accordingly, the relevance of the information provided, relative to the action being performed, appears to be vital in driving sprint and jump performance in young individuals.

There are some limitations to this study. The terms "jump as high as you can" and "sprint as fast as you can" were neutral cues that were used as controls against which the various ICs, ECs and ADCs were compared [23, 24]. This was based on previous research [23, 24] but, on the whole, the experimental terms were not more effective than these neutral control cues. It is possible that this was because there is no established standard as to what constitutes a "control cue" meaning these neutral cues were just as effective as the experimental cues in driving performance in the study participants. Similarly, the ECs and ICs required the participants to retain a specific focus for performance whereas the neutral cues simply requested maximal performance. This small differential could impact on an individual's understanding of a particular cue and though it was deliberate in nature, researchers must work to standardise cues across various tasks to ensure the most effective form of communication. Accordingly, alternative cues with different compositions, and in other languages, should be tested to examine the most effective coaching terms to enhance performance in young individuals. An important consideration here is for researchers to compare the effects of content-matched cues in both naïve and non-naïve populations alike as the results could be different in each based on different combinations of the type of cue delivered and the level of experience of the performer. A further related issue could relate to the competitive level of the youths recruited to this study.

None of the cohorts could truly be classified as "elite" and on that basis, may not have been exposed to the duration of high-quality training that might serve as a platform to leverage such a nuanced coaching technique. Future studies should consider these techniques in elite youth performers with multiple years of training experience which could differentiate them from the participants in the current study. Parallel research should also investigate the potential underlying mechanisms that determine attentional focus during motor skill execution in youth performers.

## Conclusion

The results of this study did not generally indicate that the type of experimental cue used had a positive effect on performance tests such as vertical jump and 20 m sprint in youths. There was no evidence to support the initial hypotheses that ECs, ICs and ADCs would be more effective at enhancing vertical jump and 20 m sprint performance than neutral control cues, or that ECs would be more effective than ICs for the same measures. There was, however, some limited evidence that ADCs were more effective than both ECs and ICs at enhancing 20 m sprint performance.

Based on these results, ECs, ICs, and ADCs do not seem to positively affect vertical jump or 20m sprint performance in youths. However, the findings should not deter practitioners from using such cues in youth populations as such strategies can still help with the learning and performance of a motor skill. Indeed, despite these results, it is important to note that the experimental cues did not appear to impede performance and so, at the current time, there is no reason to suggest that ECs, ICs and ADCs should not be used in different contexts and with different populations.

Practitioners are encouraged to consider that the level of experience, contextual understanding and attentional capacity of youths could be several of the reasons as to why the current results deviate from accepted empirical evidence that suggests that the provision of ECs, and possibly ADCs, can result in enhanced motor performance. Future research must be undertaken to confirm the influence of the aforementioned factors and should investigate how ECs, ICs, and ADCs affect the performance and learning of motor skills by including more motor skill tasks and extensive measures in studies.

## Supporting information

**S1 Data. Boarding school data.**
(XLSX)

**S2 Data. Grammar school data.**
(XLSX)

**S3 Data. Iran data.**
(XLSX)

**S4 Data. Rugby data.**
(XLSX)

**S5 Data. Soccer data.**
(XLSX)

**S6 Data. Tunisia data.**
(XLSX)

**S1 File. Cue order schemes.**
(DOCX)

**S2 File. French, Persian cues.**
(DOCX)

**S1 Fig. Jump forest plots.**
(TIF)

**S2 Fig. Sprint forest plots.**
(TIF)

## Author Contributions

**Conceptualization:** Jason Moran, Matt Allen, Norodin Vali, Mike Davies, James Earle, Megan Klabunde, Gavin Sandercock.

**Data curation:** Jason Moran, Raouf Hammami, Joshua Butson, Abdelkader Mahmoudi, Norodin Vali, Ieuan Lewis, Phil Samuel, Mike Davies, James Earle, Megan Klabunde, Gavin Sandercock.

**Formal analysis:** Jason Moran, Joshua Butson, Matt Allen, Abdelkader Mahmoudi, Norodin Vali, Ieuan Lewis, James Earle, Megan Klabunde, Gavin Sandercock.

**Investigation:** Jason Moran, Joshua Butson, Matt Allen, Abdelkader Mahmoudi, Norodin Vali, Ieuan Lewis, Phil Samuel, Mike Davies, James Earle, Megan Klabunde, Gavin Sandercock.

**Methodology:** Jason Moran, Raouf Hammami, Joshua Butson, Matt Allen, Abdelkader Mahmoudi, Norodin Vali, Ieuan Lewis, Mike Davies, James Earle, Megan Klabunde, Gavin Sandercock.

**Supervision:** Jason Moran.

**Writing – original draft:** Jason Moran, James Earle, Megan Klabunde, Gavin Sandercock.

**Writing – review & editing:** Jason Moran, James Earle, Megan Klabunde, Gavin Sandercock.

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
