## [Decision Letter · Decision Letter 0]

28 Aug 2022

PONE-D-22-21775Do verbal coaching cues and analogies affect motor skill performance in youth populations?PLOS ONE

Dear Dr. Moran,

Thank you for submitting your manuscript to PLOS ONE. After careful consideration, we feel that it has merit but does not fully meet PLOS ONE’s publication criteria as it currently stands. Therefore, we invite you to submit a revised version of the manuscript that addresses the points raised during the review process.

Both reviewers agreed that the article has merit and after major corrections can be acceptable for publication due to the quality of the approach. Please ensure that your decision is justified on PLOS ONE’s publication criteria and not, for example, on novelty or perceived impact.

We look forward to receiving your revised manuscript.

Kind regards,

Filipe Manuel Clemente, PhD

Academic Editor

PLOS ONE

Journal Requirements:

3. During our internal checks, the in-house editorial staff noted that you conducted research or obtained samples in another country. Please check the relevant national regulations and laws applying to foreign researchers and state whether you obtained the required permits and approvals. Please address this in your ethics statement in both the manuscript and submission information.

4. Please include a complete copy of PLOS’ questionnaire on inclusivity in global research in your revised manuscript. Our policy for research in this area aims to improve transparency in the reporting of research performed outside of researchers’ own country or community. The policy applies to researchers who have travelled to a different country to conduct research, research with Indigenous populations or their lands, and research on cultural artefacts. The questionnaire can also be requested at the journal’s discretion for any other submissions, even if these conditions are not met.  Please find more information on the policy and a link to download a blank copy of the questionnaire here: https://journals.plos.org/plosone/s/best-practices-in-research-reporting. Please upload a completed version of your questionnaire as Supporting Information when you resubmit your manuscript.

7. Please amend your authorship list in your manuscript file to include author Phil Samuel.

8. Please include your full ethics statement in the ‘Methods’ section of your manuscript file. In your statement, please include the full name of the IRB or ethics committee who approved or waived your study, as well as whether or not you obtained informed written or verbal consent. If consent was waived for your study, please include this information in your statement as well. 

9. Please include your tables as part of your main manuscript and remove the individual files. Please note that supplementary tables (should remain/ be uploaded) as separate "supporting information" files.

10. Please include captions for your Supporting Information files at the end of your manuscript, and update any in-text citations to match accordingly. Please see our Supporting Information guidelines for more information: http://journals.plos.org/plosone/s/supporting-information. 

Additional Editor Comments:

Both reviewers agreed that the article has merit and after major corrections can be acceptable for publication due to the quality of the approach.

Reviewers' comments:

Reviewer's Responses to Questions

**Comments to the Author**

1. Is the manuscript technically sound, and do the data support the conclusions?

Reviewer #1: Partly

Reviewer #2: Partly

2. Has the statistical analysis been performed appropriately and rigorously? 

Reviewer #1: Yes

Reviewer #2: No

3. Have the authors made all data underlying the findings in their manuscript fully available?

Reviewer #1: Yes

Reviewer #2: Yes

4. Is the manuscript presented in an intelligible fashion and written in standard English?

Reviewer #1: Yes

Reviewer #2: Yes

5. Review Comments to the Author

Reviewer #1: General comments:

Title

Are presented satisfactorily.

Abstract

It is written in a structured way, however, the methodology is written in a very summarized way which ends up making the findings and conclusions of the article.

There is no description of the methodology in the abstract that would allow a better understanding of what was researched.

The conclusions are very timid, making the main findings difficult.

Please confirm that the Keywords are listed as descriptors in health sciences.

Introduction

This is very extensive, and on the other hand methodologically explains some points that should this in methodology and not in the introduction. The introduction is not starting from general to specific. It should initially present a more general approach and gradually address the problem (gap) and then present the objective.

The introduction should be more focused on the construct and not on the methodology of what is being researched.

Mentioning that there are few studies or that research is scarce does not seem to me to be a robust justification for the study. please review this.

Methods

It should present more clearly the design of the study. A CONSORT or time line should be presented in order to get a better view of the study design.

The sample should be better explained with the number of subjects presented initially and then present the inclusion and exclusion criteria.

Results

I consider that the results were very simplistic and practically do not clarify what was proposed. I suggest that this topic be further explored in order to better support the manuscript.

Discussion

It should reaffirm the objectives and start discussing the results in the chronological order that appear in the item results.

Conclusion

Are presented satisfactorily.

References

Please confirm the formatting of the references and of the 37 references 19 are current and 18 are more than five years old. It is suggested that the references be updated.

Overview

The manuscript presented addresses a relevant research topic.

It would be advisable to do a general review.

Reviewer #2: Review of Manuscript: PONE-D-22-21775 Do verbal coaching cues and analogies affect motor skill performance in youth

populations?

Title: Do verbal coaching cues and analogies affect motor skill performance in youth

populations?

The authors have written a manuscript investigating cues of external (EC) or internal (IC) focus, with an additional condition of analogies with a directional component (ADC) on sprint and jump performance in youth performers. A control cue condition was also included.

The authors have designed an interesting study of the attentional focus/cueing literature. However, it is recommended that the authors reorganize this original manuscript to improve the clarity and understanding of both the findings and rationale for the study.

Below are general comments for the author's consideration.

General Comments (GC):

GC1: There are no lines or pages provided so specific comments will attempt to direct the author by sections, paragraphs, and sentence order. For future submissions, authors should consider including page and line numbers.

GC2: Authors are encouraged to include what measures were chosen to indicate ‘performance’ and ‘retention’ in the abstract. Currently, it is ambiguous and the abstract does not indicate what is measured, i.e. sprint performance (time, velocity, etc) and jump performance (distance/height, etc).

GC3: Many of the statements are written colloquially. The authors are encouraged to stick to an objective form of writing and reduce any colloquial/subjective overtones.

GC4: The authors are encouraged to either choose ‘attentional focus’ or ‘cues’ as the nomenclature for delivering relevant information. Alternatively, discuss how they are related. It is true that cues can direct the focus of an athlete, but the reader needs to have background knowledge of what a cue is. Currently, the author introduces attentional focus and then switches to the nomenclature of cues. Similarly, ‘instructions’ are also discussed. Authors are encouraged to choose either ‘cues’, ‘attentional focus’, or ‘instructions’.

GC5: A concern of neuromuscular fatigue for these athletes is worrying. Each participant spent about 25 to 30 minutes performing either a 20-m sprint or “jump”. Is there any evidence to support that 10 repetitions of 20-metre sprints and jumps would not be affected by neurological fatigue? Particularly populations are not homogeneous in their training experience.

Below are specific comments for the author's consideration.

Specific Comments:

Introduction

Paragraph 1, Sentence 1: Authors are encouraged to change the wording so it is unique when compared to the abstract.

Paragraph 1: The paragraph seems rushed, without defining or describing concepts properly. It begins with talking about ‘learning’ and then finishes with discussing ‘performance’. From a motor control perspective, these two concepts are noticeably different. Similarly, although the constrained action hypothesis outcome is described, it is unclear how it underpins the automaticity of movement. The authors are encouraged to describe the importance of each of these concepts and how they are related. Currently, the reader needs to be aware of attentional focus (AF) literature to make sense of the paragraph.

Paragraph 2: The paragraph needs to be rewritten. The authors being concerned about the lack of research on how AF can affect fundamental movement skills (FMS) is an opinion and not objective. Stick to the facts, and report what the literature supports and what are the gaps. Refrain from including any opinions. Furthermore, it could be argued that sport-specific skills require more physical literacy than FMS, as FMS are gross-motor tasks whereas sport-specific skills require fine-motor coordination, thus performers need to be more physically literate to perform sport-specific tasks. Lastly, without supporting evidence that physical education (PE) should be teaching FMS over sport-specific skills, this comes across as a subjective perspective of the author. PE classes should be progressing into sport-specific skills after FMS have been taught, yet it could be true that FMS should not be overlooked.

Paragraph 3, Sentence 1: Authors are encouraged to rewrite the sentence for clarity.

Paragraph 3, Sentence 3-4: Authors need to include a reference for these claims.

Paragraph 4, Sentence 2: GC3

Paragraph 4: The authors are encouraged to either define analogies or metaphors. Also, if they are the same, stick to one term and use it throughout the manuscript. Authors are also encouraged to indicate how they can be different to instructions/AF.

Paragraph 4, Last Sentence: The authors are encouraged to rewire the sentence for clarity. A distal focus is done by focusing distally from the body, not by it simply being an EC or an analogy. For example, throwing a dart and focusing on the dart and or board are both ECs, but one is more distal than the other. Similarly, focusing on throwing the dart quickly like a gunshot or arrow, are both analogies, but do not give proximal or distal information.

Paragraph 5, Sentence 2: Change “our” to “the authors” as it changes it from a subjective (2nd person) perspective to an objective (3rd person) perspective.

Paragraph 5: The authors are encouraged to change the names of the conditions to reflect the specificity of the attentional focus, i.e. the “directional component” is drawing the attention externally in both ‘toward’ and ‘away’ conditions.

Paragraph 5, Sentence 4-5: GC3

Methods

Paragraph 1: GC3. Authors are also recommended to include the total amount of sprints and jumps performed per participant per testing session.

Paragraph 2: The authors are encouraged to be more explicit as to the data processing. Currently, it is interpreted that a comparison across all the populations and measures was used to increase statistical power. However, these measures are open to an increase in error with the differences across training experiences, surfaces, apparatus, and matched-controlled cues. The mean of a highly trained group will be different than a moderately or poorly trained group, why are these means collated together and not interpreted separately?

Paragraph 3: The authors are encouraged to rewrite the paragraph for clarity. The notion that the control cues provided no attentional component is false. The attention was to either jump HIGH or sprint FAST. No attentional component would simply be, “perform a jump” or “perform a sprint”. The current control conditions are of neutral focus, as they do not direct the attention internally or externally, but still direct the attention of the athlete.

Paragraph 4 (Warm-Up): The authors are encouraged to describe if any feedback, i.e. cues or AF was given during the warm-up.

Paragraph 5 (Jumps): The authors are encouraged to rewrite the paragraph for clarity. Is this a countermovement jump? It can be inferred, but authors are encouraged to explicitly tell the reader what type of jump was completed. Similarly, the order in which ‘cues’, ‘instructions’ or ‘feedback’ was given is confusing. It seems that “jump as high as you can” (which is the control condition) was a higher dosage than any other condition. Why? Were these jumps performed Akimbo? Why or why not? How was the data then processed? Best trial on what measure? An average across two jumps? The ‘highest’ of the two trials is unclear, based on what measure, jump height?

Paragraph 6 (Sprints): The authors are encouraged to rewrite the paragraph for clarity. Similar to the jumps paragraph, the order in which ‘cues’, ‘instructions’ or ‘feedback’ was given is confusing. It seems that the participants had to focus on a lot of information prior to performing the sprint, however, the ‘control’ condition relevant information was constantly reinforced throughout each condition?

Results

The results are reported succinctly and clearly.

Discussion

Paragraph 1, Sentence 3-4: The authors are encouraged to combine these sentences to increase the clairt and reduce leading the reader to believe more evidence than a very specific finding is supporting evidence.

Paragraph 2: The authors are encouraged to include the maturity levels of the participants to strengthen this discussion point. The comprehension of the verbal cue is important and does develop with an increase in maturity. However, it is unclear what the maturity age of the participants is as only chronological age is included. Were the participants, pre-, circa-, or post-peak height velocity?

Paragraph 2, Sentence 4-5: The authors are encouraged to rewrite the sentences for clarity. Sentence 4 is also a run-on.

Paragraph 3, Sentence 1-3: The authors are encouraged to put this content in the introduction as it introduces the relevance of the constrained action hypothesis to the reader.

Paragraph 3: The notion that the constrained action hypothesis does not work in children is not supported by the literature. See the following:

Tse ACY, van Ginneken WF. Children’s conscious control propensity moderates the role of attentional focus in motor skill acquisition. Psychol Sport Exerc 31: 35–39, 2017.

Chow JY, Koh M, Davids K, Button C, Rein R. Effects of different instructional constraints on task performance and emergence of coordination in children. Eur J Sport Sci 14: 224–232, 2014.

Prapavessis H, McNair PJ, Anderson K, Hohepa M. Decreasing landing forces in children: The effect of instructions. J Orthop Sports Phys Ther 33: 204–207, 2003.

Paragraph 4: The authors are considering the participants as naïve participants, but some have training ages of 4+ years. Could it instead of an effect of training age rather than all young participants being considered naïve? Young participants have been shown to elicit high levels of force and power production as long as they are trained appropriately. See:

Lesinski M, Prieske O, Granacher U. Effects and dose-response relationships of resistance training on physical performance in youth athletes: A systematic review and meta-analysis. Br J Sports Med 50: 781–795, 2016.

Paragraph 5: The authors are eluding that the information may not have been relevant enough. However, Winkelman also eludes that when cues are compared, they should be matched-controlled (see Winkelman NC, Clark KP, Ryan LJ. Experience level influences the effect of attentional focus on sprint performance. Hum Mov Sci 52: 84–95, 2017.) Thus, it is possible that the neutral cue, which was reinforced throughout each condition across jump and sprint tests received more exposure and was also simpler and more similar to what a coach would say than the other ‘cues’ that were more instructions rather than a practical cue.

Paragraph 6: See comments for paragraph 5 and reference that these ‘control’ cues still provided an attentional focus, to either jump HIGH or sprint FAST. Thus, they were not controls, but rather neutral focused cues. Limitations should rather point to the lack of matched cues, homogenous training age, language, or maturity level. All of which were described in the previous discussion paragraphs.

Conclusion

The authors should consider re-writing the conclusion to include the considerations of the reviewer. The study is interesting, but the way the manuscript is written indicates that external cues do not provide benefit over internal or neutral cues in young populations. This notion contest previous research and should be given a deeper critical evaluation as to why.

6. PLOS authors have the option to publish the peer review history of their article (what does this mean?). If published, this will include your full peer review and any attached files.

Reviewer #1: **Yes: **Felipe J. Aidar

Reviewer #2: **Yes: **Saldiam R Barillas

---

## [Author Response · Author response to Decision Letter 0]

27 Sep 2022

We would like to sincerely thank both reviewers and editor for their time and efforts in helping to improve our manuscript. We feel the recommended changes have improved the paper, particularly in relation to its scientific rigour and replicability. Below, is an itemised response to each of the points raised and we hope that this brings the paper up to the high standard required for publication in PLOS One. In some cases, both reviewers requested similar changes and these were relatively simple to implement. In others, there were instructions that somewhat contradicted each other (for example the results section!) so we did our very best to come the most appropriate compromise between the recommendations of each reviewer. In places, we have subdivided down the reviewers’ comments into smaller points so as to address the specific aspects of each of the comments. We hope that in all cases that our responses reflect the requests of the reviewers and we once again thank them for their valuable input.

COMMENTS FOR REVIEWER #1

General comments:

Reviewer: Title

Are presented satisfactorily. 

Response: We appreciate this comment.

Reviewer: Abstract

It is written in a structured way, however, the methodology is written in a very summarized way which ends up making the findings and conclusions of the article. There is no description of the methodology in the abstract that would allow a better understanding of what was researched.

Response: We appreciate that our description of the methods was ambiguous here so we added more detail to specifically reflect the approach we took in a clearer way:

“Across several international locations, we undertook a series of separate experiments to determine the effect of external cues (EC), internal cues (IC), analogies with a directional component (ADC) and control cues on sprint time (20 m) and vertical jump height in youth performers. For greater statistical power, we combined these data using internal meta-analytical techniques to pool results across each test location. We amalgamated this approach with a repeated measures analysis to determine if there were any differences between the ECs, ICs and ADCs within the different experiments.”

Because we have used a novel statistical approach to pool data we felt the description of the methods needed to heavily reflect the statistical approach taken. To indicate to the reader the general objective we wrote:

“Across several international locations, we undertook a series of separate experiments to determine the effect of external cues (EC), internal cues (IC), analogies with a directional component (ADC) and control cues on sprint time (20 m) and vertical jump height in youth performers.”

We would also hope that the state purpose of the research would also give the reader some insight into the reason we undertook it. However, in response to the reviewer’s comment we also slightly edited the ‘purpose’ section of the abstract:

“The way coaching instructions are worded to performers can impact on the quality with which a subsequent motor skill is executed. However, there have been few investigations on the effect of coaching instructions on basic motor skill performance in youths.”

Reviewer: The conclusions are very timid, making the main findings difficult.

Response: Given that our results were somewhat controversial in this area, we wanted to be equivocal about their implications for youths. However, we believe the reviewer is correct and we should not shy away from our otherwise interesting findings. Accordingly, we have now written:

“These results suggest that the type of cue or analogy provided to a youth performer has little subsequent effect on sprint or jump performance. Accordingly, coaches might take a more specific approach that is suited to the level or preferences of a particular individual.”

Reviewer: Please confirm that the Keywords are listed as descriptors in health sciences.

Response: We have now given the keywords a more health-orientated focus:

“Fundamental movement skills, instructions, run, jump, youth”

Introduction

Reviewer: This is very extensive, and on the other hand methodologically explains some points that should this in methodology and not in the introduction.

Response: Please see corresponding answer below for explanation on this particular request by the reviewer.

Reviewer: The introduction is not starting from general to specific. It should initially present a more general approach and gradually address the problem (gap) and then present the objective.

Response: We agree with the reviewer that we were too quick to get into the main details of the article without first explaining to readers about the underpinning theory and giving some relevant background. We have now rearranged the introduction to first explain what internal and external cues are and then to relate them to the constrained action hypothesis which is the general theory that underpins research of this type. Only after this do we delve into the relevant research and the related gaps that exist:

“Coaching instructions that direct a performer’s attention externally (i.e. a focus placed outside of the body) or internally (i.e. a focus on body part movement) have been shown to have demonstrable effects on subsequent motor skill performance (1–3). The constrained action hypothesis (6) suggests that an external focus of attention can result in improved motor performance by increasing the automaticity of movement control during a given action, also supporting implicit learning (7).”

We have also moved the evidence cited by Winkelman towards the beginning of the introduction as this was more general in nature and was better placed earlier in the work to set the scene for the reader, as per the reviewer’s request.

Reviewer: The introduction should be more focused on the construct and not on the methodology of what is being researched.

Response: With the changes we have now made, we hope that this particular request of the reviewer has been addressed in the process. As per our answer about the methods section above, we believe a certain amount of information on this was required in the introduction section as some of the concepts have not been studied before and could be unfamiliar even to researchers and practitioners who are very familiar with the literature in this area. As our study is the first of its kind in this population, and with the described methods, we believe some relevant background information was necessary to provide early in the piece. We hope the reviewer can appreciate our approach on this.

Reviewer: Mentioning that there are few studies or that research is scarce does not seem to me to be a robust justification for the study. please review this.

Response: Thank you for this observation – we agree that more justification for the study is required. In addition to the changes detailed below, we believe that in too many places we highlighted the lack of research in this area and we have now reduced this in line with the reviewer’s request. We have now added a sentence which justifies exactly why this scarce research has negative implications for coaching and learning in a practical context. We hope this adds further justification as to why the low amount of research in this area has further effects on coaching and learning:

“To date, there have been very few investigations on the effect of various different foci of attention on the performance or retention of basic FMS, such as running and jumping, in youths. This has made it particularly difficult to determine whether or not the manipulation of attentional focus can enhance performance and subsequent motor learning in young individuals. Accordingly, coaches and teachers may therefore not be using optimal methods when attempting to drive motor learning in these populations.”

We would also direct the reviewer to the pre-existing text that is below this additional text which also describes the problems associated with the low amount of studies in this area:

“Two systematic reviews (6,7) have provided a comprehensive overview of the various studies that have been undertaken in this area, however, of the investigations carried out in children, most related to sport-specific skills such as basketball dribbling, golf putting and tennis serving, as opposed to basic FMS that might be taught in physical education, such as jumping. Moreover, not a single study in children has examined the effect of different attentional foci on the execution of running or sprinting, a vital FMS for children as they develop. This is particularly concerning given that physical literacy has been associated with higher levels of physical activity (8) meaning that if youths can achieve mastery in FMS such as running, they may be more likely to engage in physical activity across the lifespan (9). Accordingly, the teaching of these skills in a way that best resonates with the target population appears to be a vital component of physical education provision.”

Further down in the introduction, we have also added more justification alongside the initial observation that research in the area is scarce:

“Moreover, to our knowledge, no previous study has examined these types of foci when combined with analogies in any population, let alone in youths; this despite the merging of ECs with analogies being suggested to be an effective way to retain a distal focus for optimal performance (1,4). This could have important implications for coaching and learning as the combination of ECs and analogies could represent a new tool for coaches that can enhance the contextual understanding of a performer and, by extension, the learning of key movement skills.”

Methods

Reviewer: It should present more clearly the design of the study. A CONSORT or time line should be presented in order to get a better view of the study design.

Response: We have now included a diagram to depict the research design of the study. We have included it as Figure 1 in the text.

Reviewer: The sample should be better explained with the number of subjects presented initially and then present the inclusion and exclusion criteria.

Response: We have now added further information to this effect in the ‘Experimental design’ section of the methods:

“We recruited 173 participants from a variety of different backgrounds, developmental levels and ages. The descriptive characteristics of the various groups can be viewed in Table 1. Only individuals under the age of 18 were eligible to take part and though the study was open to female participants, we were unable to recruit any into the various cohorts. Only healthy individuals were considered and the study was open to both trained and untrained participants.”

Results

Reviewer: I consider that the results were very simplistic and practically do not clarify what was proposed. I suggest that this topic be further explored in order to better support the manuscript.

Response: We were initially restricted by space for figures and tables as indicated in the author guidelines and so we had to include the forest plots (which contain the relevant information) in the supplemental material. However, in line with the reviewer’s request, we have now added further information to the results section. The most important change relates to the internal meta-analytical findings. For example, the reader was previously referred to the supplemental information to view the forest plots, however, they no longer need to do this as we have summarized the results in the text:

“In summary, there were no significant differences between the control cue and coaching instructions in any of the analyses. Effect sizes, representing the differences between performance under the various instructions, ranged from -0.03 (95% confidence interval: -0.24, 0.18) to 0.07 (95% confidence interval: -0.14, 0.28) for the sprint measures and were all classified as ‘trivial’. Effect sizes for the jump measures ranged from -0.30 (95% confidence interval: -0.54, -0.05) to -0.15 (95% confidence interval: -0.40, 0.09). Across all analyses, the only effect size that crossed the threshold from ‘trivial’ to ‘small’ was that for the IC (“as you jump, focus on extending your legs”) when compared to the control cue ("jump as high as you can") in the jump analysis. In that instance, there was a ‘small’ statistically significant effect size which favoured the control cue (d = -0.30, 95% confidence interval: [-0.54, -0.05], p=0.02).”

In relation to the above, we previously had only reported one effect size that had crossed the threshold from ‘trivial’ to ‘small’, however, we had failed to indicate whether this effect size related to the jump analysis or to the sprint analysis. We have now made this clear in the text. We have also added a p-value for this statistic. To help the readers to understand this result, we have also added in brackets, the coaching cues that relate to the particular results (i.e. “as you jump, focus on extending your legs”) so that context is provided while reading this section.

In relation to the second paragraph of the results which relates to the repeated measures analysis, we left this as it was because all relevant p-values and statistics are already contained in Table 3 which is in the text and is not supplementary material. In the interests of avoiding repetition, we refer the reader to the table instead of repeating the results in the text. We hope the reviewer can appreciate our rationale for this decision. We have, however, clarified and edited some of the text in this paragraph to demonstrate the results to the reader in a more understandable way.

Discussion

Reviewer: It should reaffirm the objectives and start discussing the results in the chronological order that appear in the item results.

Response: We have now started the discussion by saying:

“The purpose of this research was to determine the effectiveness of ICs, ECs and ADCs on motor skill performance in youths in various different populations, ranging from school children to academy athletes, and across a variety of international contexts and languages. We hypothesised that ECs, ICs and ADCs would be more effective than control cues for increasing jump and sprint performance, that ECs would be more effective than ICs and that ADCs would be more effective than both ECs and ICs.”

We had initially stated the results as they appeared in the results section but we appreciate that this may not have been clear enough and we have added some additional wording at the start of the discussion to differentiate the results of the two main analyses. As requested by the reviewer, the meta-analytical results are discussed first in the discussion as they are listed first in the results and the repeated measures second.

Conclusion

Reviewer: Are presented satisfactorily. 

Response: We would like to thank the reviewer for this comment.

References

Reviewer: Please confirm the formatting of the references and of the 37 references 19 are current and 18 are more than five years old. It is suggested that the references be updated.

Reviewer: The four main academic co-authors of this article wrote it based on what we saw as the most relevant articles on this topic. Studies such as those carried out by Wulf, Porter and Winkelman are older pieces of work but were some of the formative investigations in this área and we feel it is very necessary to retain them in our own study, despite their advancing age. We did review the reference list but feel it is appropriate and that it might be difficult to construct the paper in the way that we have if we were to change the composition of the references. We hope the reviewer can appreciate our view on this particular issue.

COMMENTS FOR REVIEWER #2

We would like to sincerely thank both reviewers and editor for their time and efforts in helping to improve our manuscript. We feel the recommended changes have improved the paper, particularly in relation to its scientific rigour and replicability. Below, is an itemised response to each of the points raised and we hope that this brings the paper up to the high standard required for publication in PLOS One. In some cases, both reviewers requested similar changes and these were relatively simple to implement. In others, there were instructions that somewhat contradicted each other (for example the results section!) so we did our very best to come the most appropriate compromise between the recommendations of each reviewer. In places, we have subdivided down the reviewers’ comments into smaller points so as to address the specific aspects of each of the comments. We hope that in all cases that our responses reflect the requests of the reviewers and we once again thank them for their valuable input.

Reviewer: The authors have written a manuscript investigating cues of external (EC) or internal (IC) focus, with an additional condition of analogies with a directional component (ADC) on sprint and jump performance in youth performers. A control cue condition was also included. The authors have designed an interesting study of the attentional focus/cueing literature. However, it is recommended that the authors reorganize this original manuscript to improve the clarity and understanding of both the findings and rationale for the study. Below are general comments for the author's consideration.

Response: Many thanks for these comments. We have made an itemised response to these comments below.

General Comments (GC):

Reviewer: GC1: There are no lines or pages provided so specific comments will attempt to direct the author by sections, paragraphs, and sentence order. For future submissions, authors should consider including page and line numbers.

Response: This was an oversight by us but we have now added page and line numbers so that any further changes can be more accurately indicated. We appreciate the reviewer’s patience on this matter.

Reviewer: GC2: Authors are encouraged to include what measures were chosen to indicate ‘performance’ and ‘retention’ in the abstract. Currently, it is ambiguous and the abstract does not indicate what is measured, i.e. sprint performance (time, velocity, etc) and jump performance (distance/height, etc).

Response: We have removed the word ‘retention’ from the abstract as this was not specifically measured. We have also added edited the wording to reflect the specific measures used:

“Across several international locations, we undertook a series of separate experiments to determine the effect of external cues (EC), internal cues (IC), analogies with a directional component (ADC) and control cues on sprint time (20 m) and vertical jump height in youth performers.”

Reviewer: GC3: Many of the statements are written colloquially. The authors are encouraged to stick to an objective form of writing and reduce any colloquial/subjective overtones.

Response: We have done several general reviews of the manuscript to amend this issue. We now hope that the paper reads in a more objective manner.

Reviewer: GC4: The authors are encouraged to either choose ‘attentional focus’ or ‘cues’ as the nomenclature for delivering relevant information. Alternatively, discuss how they are related. It is true that cues can direct the focus of an athlete, but the reader needs to have background knowledge of what a cue is. Currently, the author introduces attentional focus and then switches to the nomenclature of cues. Similarly, ‘instructions’ are also discussed. Authors are encouraged to choose either ‘cues’, ‘attentional focus’, or ‘instructions’.

Response: In line with the reviewer’s suggestion, we have decided that the most appropriate option is to use the term ‘coaching cue’ throughout the work as this is used most commonly in the literature and was the term that we asked the participants to focus on in each experiment (i.e. “Focus as hard as you can on this cue during the jump”). In places we had used these terms interchangeably and we agree with the reviewer that this is not correct. In limited places we have elected to retain some of the original wording but only in an attempt to contextualise the use of ‘coaching instructions’ ‘attentional focus’. They are no longer used interchangeably in the text and we are now firmly using the term ‘cue’. 

Reviewer: GC5: A concern of neuromuscular fatigue for these athletes is worrying. Each participant spent about 25 to 30 minutes performing either a 20-m sprint or “jump”. Is there any evidence to support that 10 repetitions of 20-metre sprints and jumps would not be affected by neurological fatigue? Particularly populations are not homogeneous in their training experience.

Response: We were also aware that fatigue could have been a factor and this is why we took several controls to prevent its effects. Of course, in any environment, fatigue cannot be entirely minimised or discounted but we undertook the following stringent measures to prevent its effects on our results (all are detailed in the methods section of the paper with reference to offsetting fatigue and order effects):

Most importantly, we incorporated specific study design elements to reduce the order effects of fatigue. We detail this in the methods section entitled ‘coaching cues’:

“A Latin square design was used to offset order effects due to fatigue or other factors that could impact upon participants’ performance. Each participant was randomly allocated a specific “order scheme” (between 1 and 10) via a random number generator (https://www.random.org/). This ‘order scheme’ determined the sequence in which each individual received their instructional cues prior to jumping or sprinting with each letter corresponding to a particular coaching cue, which can be seen in the supplemental information. As participants sprinted and jumped twice for each cue, each cue appeared twice. The order schemes can also be seen in the supplemental information.”

If the reviewer refers to the supplemental information file ‘Suppl info - Cue order schemes’ they will see more information on this. In short, each participant did the sprints and jumps in different randomised orders. For example, because these orders were randomised, Participant “A” could have done the control cue fifth while Participant “B” may have done it seventh, or ninth etc etc. Simialrly, Participant “A” could have done the external cue second while Participant “B” could have done it seventh. On paper this seems rather simple but in practice was extremely challenging as each participant had to be allocated an order scheme and given their cues in a different order corresponding to that particular scheme. In the end, very few participants will have done the sprints in the same order because they were all allocated different order schemes meaning the effects of fatigue will have been mostly minimised. 

In addition to the above, we took the usual measures one would be expected to do to allow recovery of the creatine phosphate system between short quick bursts of action. In this way, every sprint and jump was theoretically a fully rested new effort done in a primed and freshened state. The following studies support our approach in minimising fatigue in this way:

Grant et al. (2011, JSCR) showed that “a 1:10 sprint-to-rest ratio allows full performance recovery between 15-m sprints”. However, in our work we used a 20 m distance but we also utilised a 1:40 sprint to rest ratio to ensure maximal recovery.

Read et al. (2001, JSCR) showed that “a 15-second rest interval was sufficient for recovery during the performance of depth jumps”. However, in our work we provided x8 times this amount with at least a two minute rest between jumps of a lower intensity.

Woolf et al (2018, ACSM) showed that 60 secs was also sufficient between efforts.

We hope the above provisions reassure the reviewer that we took utmost care to ensure that data were collected in a rigourous manner.

Specific Comments:

Introduction

Reviewer: Paragraph 1, Sentence 1: Authors are encouraged to change the wording so it is unique when compared to the abstract.

Response: This has now been edited substantially due to this and other recommendations from Reviewer #1.

Reviewer: Paragraph 1: The paragraph seems rushed, without defining or describing concepts properly. It begins with talking about ‘learning’ and then finishes with discussing ‘performance’. From a motor control perspective, these two concepts are noticeably different. Similarly, although the constrained action hypothesis outcome is described, it is unclear how it underpins the automaticity of movement. The authors are encouraged to describe the importance of each of these concepts and how they are related. Currently, the reader needs to be aware of attentional focus (AF) literature to make sense of the paragraph.

Response: We agree with the reviewer and, indeed, Reviewer #1 also made similar comments. In line with both, we have now begun the introduction in a more general way, looking to become more specific as it continues. Now, instead of proceeding straight into potentially abstract topics for a reader, we start with a definition of a coaching cue followed by an explanation of the constrained action hypothesis and its mechanisms as a sound basis for later expansion in the introduction. To achieve this we have added additional words on mechanisms from Kal et al. (2013):

“A coaching cue is a verbal instruction that can be used to focus an individual’s attention on a movement so as to optimise its execution (1). Cues that direct a performer’s attention externally (i.e. a focus placed outside of the body) or internally (i.e. a focus on body part movement) have been shown to have demonstrable effects on subsequent motor skill performance (2–4). The constrained action hypothesis (5) suggests that an external focus of attention can result in improved motor performance by increasing the automaticity of movement control during a given action, also supporting implicit learning (6). It has been proposed that an external focus reduces the attentional capacity that is needed to carry out a movement which can also be enhanced through greater coordination between working muscles (7). Accordingly, the manner in which coaching cues are worded and presented to performers can immediately impact on the quality with which motor skills are executed (8) and in the longer term, this can be reinforced through the learning process (9).

Reviewer: Paragraph 2: The paragraph needs to be rewritten. The authors being concerned about the lack of research on how AF can affect fundamental movement skills (FMS) is an opinion and not objective. Stick to the facts, and report what the literature supports and what are the gaps. Refrain from including any opinions. 

Response: Agreed. We have now removed the references to FMS and have referred only to the specific skills on which the research related to. We have made various edits throughout in accordance with this request from the reviewer.

Reviewer: Furthermore, it could be argued that sport-specific skills require more physical literacy than FMS, as FMS are gross-motor tasks whereas sport-specific skills require fine-motor coordination, thus performers need to be more physically literate to perform sport-specific tasks.

Response: In line with the previously mentioned change, this has now been edited/removed from the manuscript to avoid any confusion and to be more accurate.

Reviewer: Lastly, without supporting evidence that physical education (PE) should be teaching FMS over sport-specific skills, this comes across as a subjective perspective of the author. PE classes should be progressing into sport-specific skills after FMS have been taught, yet it could be true that FMS should not be overlooked.

Response: Agreed, and we have now removed the reference to physical education in line with this recommendation. In line with the previous two requests, we have made the necessary edits to maintain clarity and coherence with surrounding text.

Reviewer: Paragraph 3, Sentence 1: Authors are encouraged to rewrite the sentence for clarity.

Response: This sentence has now moved due to other amendments but has now been rewritten:

“Expanding on the above concept, and citing the work of Porter (2,10), Winkelman (11) highlighted the impact of including a directional component in a cue to enhance motor performance.”

Reviewer: Paragraph 3, Sentence 3-4: Authors need to include a reference for these claims.

Response: The relevant references have now been added.

Reviewer: Paragraph 4, Sentence 2: GC3

Response: As per our previous response (GC3), we have performed a general review of the writing style.

Reviewer: Paragraph 4: The authors are encouraged to either define analogies or metaphors. Also, if they are the same, stick to one term and use it throughout the manuscript. Authors are also encouraged to indicate how they can be different to instructions/AF.

Response: We have now added a definition (Capio et al., 2020) that indicates the difference to a normal cue.

“An analogy is a coaching instruction that conceals biomechanical cues within spoken words. It differs from a conventional instruction in that it conveys the key elements of a given movement without the need to specific reference those same elements (17).”

We now use only the terms ‘analogy’ and ‘analogies’ throughout. 

Reviewer: Paragraph 4, Last Sentence: The authors are encouraged to rewire the sentence for clarity. A distal focus is done by focusing distally from the body, not by it simply being an EC or an analogy. For example, throwing a dart and focusing on the dart and or board are both ECs, but one is more distal than the other. Similarly, focusing on throwing the dart quickly like a gunshot or arrow, are both analogies, but do not give proximal or distal information.

Response: Agreed – we have now removed the term ‘distal’ as it is somewhat redundant here.

Reviewer: Paragraph 5, Sentence 2: Change “our” to “the authors” as it changes it from a subjective (2nd person) perspective to an objective (3rd person) perspective.

Response: We have now changed this.

Reviewer: Paragraph 5: The authors are encouraged to change the names of the conditions to reflect the specificity of the attentional focus, i.e. the “directional component” is drawing the attention externally in both ‘toward’ and ‘away’ conditions.

Response: We have now added the term ‘external’ here to differentiate. 

Reviewer: Paragraph 5, Sentence 4-5: GC3

Response: In accordance with other changes, these sentences have been revised.

Methods

Reviewer: Paragraph 1: GC3. Authors are also recommended to include the total amount of sprints and jumps performed per participant per testing session.

Response: We have added this information.

Reviewer: Paragraph 2: The authors are encouraged to be more explicit as to the data processing. Currently, it is interpreted that a comparison across all the populations and measures was used to increase statistical power. However, these measures are open to an increase in error with the differences across training experiences, surfaces, apparatus, and matched-controlled cues. The mean of a highly trained group will be different than a moderately or poorly trained group, why are these means collated together and not interpreted separately?

Response: Thank you for this comment – we perhaps did not provide enough information at this stage of the paper. We did in fact also separately analyse all of the data from the different centres we just hadn’t indicated it here as we should have. It was an oversight as this constitutes 50% of our analyses for this paper and in line with a similar comment from Reviewer #1, we have added the following information:

“We combined this approach with a within-country repeated measures analysis to determine the effect of the various coaching cues on performance at each of the six centres involved in the study. The design of the research can be viewed in Figure 1 below.”

To come back to the reviewer’s original approach, we also appreciate the increase in error that accumulates across the various centres we used within this research. That is why we used a random effects analysis and standardised mean difference to offset these effects. Though they can not be entirely minimised, we felt a priori that this would still represent a logical approach given the lack of investigations in this population and felt that the internal meta-analytical approach could expedite knowledge in the area. We acknowledge the limitations both here and in the paper itself but combined with the repeated-measures analysis that compared the coaching cues at each of the venues, we also feel that this provides a strong, valuable and interesting two-pronged analysis that can be of use to practitioners in particular.

Reviewer: Paragraph 3: The authors are encouraged to rewrite the paragraph for clarity. The notion that the control cues provided no attentional component is false. The attention was to either jump HIGH or sprint FAST. No attentional component would simply be, “perform a jump” or “perform a sprint”. The current control conditions are of neutral focus, as they do not direct the attention internally or externally, but still direct the attention of the athlete.

Response: We appreciate the reviewer’s stance on this and have now deleted the line in question.

Reviewer: Paragraph 4 (Warm-Up): The authors are encouraged to describe if any feedback, i.e. cues or AF was given during the warm-up.

Response: No cues were provided during the warm up and we have now indicated this.

Reviewer: Paragraph 5 (Jumps): The authors are encouraged to rewrite the paragraph for clarity. Is this a countermovement jump? It can be inferred, but authors are encouraged to explicitly tell the reader what type of jump was completed.

Response: Yes, this was a vertical countermovement jump and we have now added/edited the text in this paragraph for clarity.

Reviewer: Similarly, the order in which ‘cues’, ‘instructions’ or ‘feedback’ was given is confusing. It seems that “jump as high as you can” (which is the control condition) was a higher dosage than any other condition. Why?

Response: We are unsure as to why the reviewer has concluded that the control condition was given in a higher dosage. As already indicated in the methods, each cue was given twice so all were administered an equal number of times:

“There were five different cues for the sprints and five different cues for the jumps meaning each participant received each cue twice.”

If the reviewer would be willing to provide further guidance on where this point of confusion is in the paper, we would be happy to revisit it.

Reviewer: Were these jumps performed Akimbo? Why or why not?

Response: Yes, they were performed akimbo to standardise execution across the various centres. We have now added this information to the paper:

“For the test, the arms were positioned akimbo (i.e. with the hands on the hips and the elbows turned outward) and the feet positioned approximately shoulder width, at a distance comfortable for the participant.”

Reviewer: How was the data then processed? Best trial on what measure? An average across two jumps? The ‘highest’ of the two trials is unclear, based on what measure, jump height?

Response: We have now added this information for clarity:

“each participant’s best effort (i.e. highest jump in cm) out of two trials was used in the analysis”

Reviewer: Paragraph 6 (Sprints): The authors are encouraged to rewrite the paragraph for clarity. Similar to the jumps paragraph, the order in which ‘cues’, ‘instructions’ or ‘feedback’ was given is confusing. It seems that the participants had to focus on a lot of information prior to performing the sprint, however, the ‘control’ condition relevant information was constantly reinforced throughout each condition?

Response: We acknowledge that there was quite a bit of information provided to the participants prior to the sprints, however, this was in an effort to stay in line with similar published studies in the area. We replicated the protocol of Winkelman et al. who we consider to have done very good and rigorous work in this area and so we wanted to align our methods with theirs. On the amount of information provided, this was not before every sprint but before the first one only and alongside a coach’s demonstration that was provided collectively, not individually (and was not repeated). The only information provided prior to the sprint was the specific cue as randomly allocated. We have now clarified and reworded this paragraph to address these issues.

Results

Reviewer: The results are reported succinctly and clearly.

Response: Thank you for this comment. We have added some additional detail based on a request from reviewer #1 though we believe the clarity is still retained.

Discussion

Reviewer: Paragraph 1, Sentence 3-4: The authors are encouraged to combine these sentences to increase the clarity and reduce leading the reader to believe more evidence than a very specific finding is supporting evidence.

Response: We have added additional detail here and rewritten the paragraph for greater clarity.

Reviewer: Paragraph 2: The authors are encouraged to include the maturity levels of the participants to strengthen this discussion point. The comprehension of the verbal cue is important and does develop with an increase in maturity. However, it is unclear what the maturity age of the participants is as only chronological age is included. Were the participants, pre-, circa-, or post-peak height velocity?

Response: We appreciate this point from the reviewer. Unfortunately, we did not collect biological maturity data as it would have been very difficult given what was a highly arduous and finely balanced data collection process on each testing day. Based on chronological age as a very loose indicator of “likely” biological maturity status, we did appear to have children in the pre-, mid- and post-peak height velocity stages of maturation. However, in this paragraph, we talk about neurocognitive capacities which do not develop linearly with physical capacities during growth and maturation and so the use of Mirwald’s maturity offset, Khamis and Roche’s predicted adult height or Tanner staging would have had limited use in supporting the points we were attempting to make here.

Reviewer: Paragraph 2, Sentence 4-5: The authors are encouraged to rewrite the sentences for clarity. Sentence 4 is also a run-on.

Response: We have split sentence 4 into two thus eliminating the run-on sentence. We have also split sentence 5 for clarity. 

Reviewer: Paragraph 3, Sentence 1-3: The authors are encouraged to put this content in the introduction as it introduces the relevance of the constrained action hypothesis to the reader.

Response: This information now also appears in the introduction in a similar guise give that both reviewers had requested changes that necessitated this. We felt a reiteration of this information was necessary in the discussion as we will potentially be explaining a set of unexpected results to somewhat naïve readers in many cases, and so wanted to re-include this information to provide context.

Reviewer: Paragraph 3: The notion that the constrained action hypothesis does not work in children is not supported by the literature. See the following:

Tse ACY, van Ginneken WF. Children’s conscious control propensity moderates the role of attentional focus in motor skill acquisition. Psychol Sport Exerc 31: 35–39, 2017.

Chow JY, Koh M, Davids K, Button C, Rein R. Effects of different instructional constraints on task performance and emergence of coordination in children. Eur J Sport Sci 14: 224–232, 2014.

Prapavessis H, McNair PJ, Anderson K, Hohepa M. Decreasing landing forces in children: The effect of instructions. J Orthop Sports Phys Ther 33: 204–207, 2003.

Response: We agree with the reviewer that our language was far too strong here and we have now moderated it according to the reviewer’s recommendations. We also acknowledge the literature cited by the reviewer and agree that the constrained action hypothesis can and does hold in children. At the same time we must acknowledge the strength of our own results and so we believe the rewriting of the text in question now offers a better balance between this and the extant literature:

“Based on the extant evidence, these concepts seem to hold in adults (37) yet according to our results at least, the predicted outcomes of constrained action hypothesis could be impeded in a youth population. Why this is the case is not entirely clear; however, there could be a practical explanation for our finding that has previously been raised by Maxwell and Masters (38). These authors demonstrated that when asked to perform a balance task, performers who were provided with an internal focus of attention had switched to an external focus of attention once they executed the task.”

Reviewer: Paragraph 4: The authors are considering the participants as naïve participants, but some have training ages of 4+ years. Could it instead of an effect of training age rather than all young participants being considered naïve? Young participants have been shown to elicit high levels of force and power production as long as they are trained appropriately. See:

Lesinski M, Prieske O, Granacher U. Effects and dose-response relationships of resistance training on physical performance in youth athletes: A systematic review and meta-analysis. Br J Sports Med 50: 781–795, 2016.

Response: Yes, the reviewer makes a good point and we do acknowledge the fact that young people can indeed exert high levels of force. To clarify, we don’t necessarily see ‘training age’ as the same thing as ‘life experience’. For example, we have a number of active coaches on our research team for this study and several of them have suggested that even their more experienced athletes can be naïve in terms of their life experiences, their ability to follow instructions and their understanding of what coach is asking. Essentially, despite being experienced trainees, they remain “immature” for want of a better term. In many cases, an experienced athlete could have undertaken four years of training but never been exposed to analogies as a coaching tool and so they would remain naïve to their use in this context. We have now clarified the text here to avoid confusing our readers. For example, we now differentiate ‘experience’ from ‘life experience’ so as to provide further context to this paragraph.

Reviewer: Paragraph 5: The authors are eluding that the information may not have been relevant enough. However, Winkelman also eludes that when cues are compared, they should be matched-controlled (see Winkelman NC, Clark KP, Ryan LJ. Experience level influences the effect of attentional focus on sprint performance. Hum Mov Sci 52: 84–95, 2017.) Thus, it is possible that the neutral cue, which was reinforced throughout each condition across jump and sprint tests received more exposure and was also simpler and more similar to what a coach would say than the other ‘cues’ that were more instructions rather than a practical cue.

Response: We do agree that the neutral cue is simpler and more similar to what a coach would say and that in itself is very interesting as perhaps this is what coaches should be doing with young people rather than using internals, externals or analogies? Our evidence isn’t quite unequivocal enough to support that claim just yet but it is a potential avenue of future study. That aside, we are a little unsure as to what the reviewer means in this instance as the neutral cue was not delivered more times that the experimental ones. As now clarified in the method section, the neutral cue was delivered the same amount of times (i.e. twice) as every other cue so would not have been more reinforced with the participants. If we are misunderstanding the reviewer here, we would be happy to revisit the comment upon further advice.

One potential reason this confusion may have been caused was the way in which we described the internal meta-analytical analysis so we have added a clarifying sentence to the statistical analysis section of the methods:

“The neutral control cue was compared separately against the ECs, ICs and two different ADCs resulting in four analyses for jumps and four for sprints (EC vs control, IC vs control, and ADCs (x2) vs. control).”

Reviewer: Paragraph 6: See comments for paragraph 5 and reference that these ‘control’ cues still provided an attentional focus, to either jump HIGH or sprint FAST. Thus, they were not controls, but rather neutral focused cues. Limitations should rather point to the lack of matched cues, homogenous training age, language, or maturity level. All of which were described in the previous discussion paragraphs.

Response: We have now added the term neutral here – electing to call the terms ‘neutral control’ cues in this instance. From a research design perspective we believe it will help with readers’ understanding of our work if it is highlighted that these neutral cues were used as the control condition because they were the condition against which all the other cues were compared. In line with the reviewer’s recommendation, we now say the below and have made similar changes throughout the manuscript (including the methods):

“The terms "jump as high as you can" and "sprint as fast as you can" were neutral cues that were used as control conditions against which the various ICs, ECs and ADCs were compared (25,26).” 

We also state:

“There is a chance that this was because there is no established standard as to what constitutes a control cue meaning these neutral cues were just as effective as the experimental cues in driving performance in the study participants.”

In relation to limitations, we have now added language and biological maturation to the text and had already referenced the fact that experience level was a potential limitation in this investigation. 

Conclusion

Reviewer: The authors should consider re-writing the conclusion to include the considerations of the reviewer. The study is interesting, but the way the manuscript is written indicates that external cues do not provide benefit over internal or neutral cues in young populations. This notion contest previous research and should be given a deeper critical evaluation as to why.

Response: We have edited the conclusion to add some critical insight as to why the external cues did not result in enhanced performance. This possibly relates to life experience and the attentional capacity of youths relative to adults and we have used this as a basis to recommend future work in this area.

---

## [Decision Letter · Decision Letter 1]

28 Oct 2022

PONE-D-22-21775R1Do verbal coaching cues and analogies affect motor skill performance in youth populations?PLOS ONE

Dear Dr. Moran,

Thank you for submitting your manuscript to PLOS ONE. After careful consideration, we feel that it has merit but does not fully meet PLOS ONE’s publication criteria as it currently stands. Therefore, we invite you to submit a revised version of the manuscript that addresses the points raised during the review process.

We look forward to receiving your revised manuscript.

Kind regards,

Filipe Manuel Clemente, PhD

Academic Editor

PLOS ONE

Journal Requirements:

Reviewers' comments:

Reviewer's Responses to Questions

**Comments to the Author**

1. If the authors have adequately addressed your comments raised in a previous round of review and you feel that this manuscript is now acceptable for publication, you may indicate that here to bypass the “Comments to the Author” section, enter your conflict of interest statement in the “Confidential to Editor” section, and submit your "Accept" recommendation.

Reviewer #2: (No Response)

2. Is the manuscript technically sound, and do the data support the conclusions?

Reviewer #2: Partly

3. Has the statistical analysis been performed appropriately and rigorously? 

Reviewer #2: Yes

4. Have the authors made all data underlying the findings in their manuscript fully available?

Reviewer #2: Yes

5. Is the manuscript presented in an intelligible fashion and written in standard English?

Reviewer #2: Yes

6. Review Comments to the Author

Reviewer #2: Review of Manuscript: PONE-D-22-21775 Do verbal coaching cues and analogies affect motor skill performance in youth

populations?

Title: Do verbal coaching cues and analogies affect motor skill performance in youth

populations?

The authors have written a manuscript investigating cues of external (EC) or internal (IC) focus, with an additional condition of analogies with a directional component (ADC) on sprint and jump performance in youth performers. A control cue condition was also included.

The authors have addressed a majority of the comments. There are still a few general comments and specific comments that would strengthen the rigour and quality of the manuscript.

Below are general comments for the author's consideration.

General Comments (GC):

GC1: The tables are not fully legible as they are cut off in the submission.

GC2: The authors are encouraged to stop using “we” and “our” in the manuscript and maintain an objective (3rd person) writing style throughout.

GC3: The discussion does little to discuss how CMJ jump height or 20m sprint performance has previously been affected by cueing interventions. Or why no benefits were observed in these particular measures. The authors are encouraged to discuss their measures.

Below are specific comments for the author's consideration.

Specific Comments:

Introduction

Line 100: Authors are encouraged to include ‘might’ or ‘may’ between ‘therefore’ and ‘not’.

Line 111-122: To strengthen the evidenced based practice angle of the paper, authors are also encouraged to review and include the following citations in this paragraph:

Radnor JM, Moeskops S, Morris SJ, Mathews TA, Kumar NT, Pullen BJ, Meyers RW, Pedley JS, Gould ZI, Oliver JL, and Lloyd RS. Developing Athletic Motor Skill Competencies in Youth. Strength Cond J 42: 54-70, 2020.

Kushner AM, Kiefer AW, Lesnick S, Faigenbaum AD, Kashikar-Zuck S, and Myer GD. Training the developing brain part II: cognitive considerations for youth instruction and feedback. Curr Sports Med Rep 14: 235-243, 2015.

Both articles indicate that analogies should be prioritised when working with younger or less trained athletes. Particularly due to the limited abstract processing skills that will develop throughout maturation and mostly in adolescents. This better describes the mechanism at work which is inferred by Fasold et al.

Line 126: Authors are encouraged to either maintain ‘toward’ and ‘away’ in quotations or no quotations. Currently, ‘towards’ is in “” whereas ‘away’ is not.

Line 129-132: Analogies combined with ECs are likely not a new tool, but rather the evidence to support its use is new. The authors are encouraged to change the wording to highlight the novelty of the understanding, not the novelty of the practice. As previous literature recommends using simpler language, analogies, and metaphors, but based on cognitive limitations and not quantified changes.

Methods

Line 201-202: As per the comment where the control cue was given more exposure than the other cues, the reviewer mistakenly read the original paragraph to read that participants were reminded to jump as high as they can with each jump. However, in the new iteration, it is now referenced here that “Prior to the jumps taking place, each participant was individually requested to ‘jump as high as you can in the remaining ten jumps’”. It is unclear, but would an athlete have experienced this format:

1. Jump as high as you can in the remaining ten jumps

2. EC / IC / ADC / Control

3. Jump 1

4. EC / IC / ADC / Control

5. Jump 2

6. EC / IC / ADC / Control

7. Jump 3

8. Etc…

If so, the neutral cue of 'Jump as high as you can' is still given a net 1 more times than the previous (this is similar to the delivery of cues for the Sprint task too). Similarly, it is delivered in a more natural format. To be a control for the IC and EC, it should have been delivered as “As you jump, focus on getting as high as you can”, rather than “Jump as high as you can”. Or alternatively, the IC be, “Extend your legs (powerfully/forcefully/quickly)” and the EC be, “Push the ground away (powerfully/forcefully/quickly)”. This mismatched control cue should be highlighted as a limitation. However, the reviewer respects that this area of research is still being defined and parameters are still being understood. Interestingly, the results point to the control cue providing the most benefit, and it’s the cue that would be most commonly used in practice and reduces cognitive processing strain with less irrelevant information.

Results

The authors have kept the succinctness of the results with the additional text. It illustrates the absence of an effect across intervention cues.

Discussion

Line 319-321: Authors are encouraged to include Kushner et al (2015) to support this statement.

Line 339-341 & Line 348 -351: The authors make a general statement that their results showcase the constrained action hypothesis may be impeded by youth populations. However, this is a very general statement and is only true for naïve youth populations with measures of akimbo CMJ jump height and 20m sprint time. The authors are encouraged to be more specific as research currently exists that young (~11-year-olds) and old (~16-year-olds) youth athletes benefit from the constrained action hypothesis (see references):

Barillas, S. R., Oliver, J. L., Lloyd, R. S., & Pedley, J. S. (2022). Kinetic Responses to External Cues Are Specific to Both the Type of Cue and Type of Exercise in Adolescent Athletes. Journal of strength and conditioning research. (published ahead of print)

Oliver, J. L., Barillas, S. R., Lloyd, R. S., Moore, I., & Pedley, J. (2021). External Cueing Influences Drop Jump Performance in Trained Young Soccer Players. Journal of strength and conditioning research, 35(6), 1700–1706.

Line 355-368: The paragraph seems to continue the argument that these young participants were unable to realise the benefits of the constrained action hypothesis. However, the argument seems to put the weighting on the participants, as they were unable to focus on only the relevant information. Rather, it could be argued that they were unable to realise the benefits of the constrained action hypothesis because they were given irrelevant information, thus blunting a potentially beneficial effect. This is supported by the neutral cue receiving the most benefit in CMJ jump height. The paragraph seems appropriate to discuss this limitation, rather it implies that only adults can benefit from external cues. Again, see the previous two references. It could be argued that if they were more trained and had more exposure to the cue to no longer be ‘naïve’ then maybe they would receive a benefit. However, in the current state, the paragraph does not read this way.

Line 370-386: This paragraph is discussing how ADCs might be an effective way of driving motor performance. However, the only two measures included in the study were CMJ jump height and 20m sprint time. What measures are included in the study to come to the conclusion that ADCs might improve motor performance? If there are measures of motor performance or perception of the cues, please include them in the methodology and results. If not, this paragraph needs to be removed as it would then be entirely speculative.

Line 388-403: The limitation paragraph briefly discusses that the lack of standardization within neutral/control cues could have led to the observed benefit. Rather, the current evidence in the field points to the neutral control cue condition having less irrelevant information, delivered in a manner more akin to a way a coach would do in practice, and potentially highlighting an extra time at the beginning of each set of jumps or sprints. It is really hard to argue that these youth populations, all of whom play sports (soccer or rugby) aside from two school groups, would not be naïve to a 20m sprint. This paragraph should take the lack of standardisation argument towards future research using content-matched cues.

Line 400-403: These lines of future research indicate that there may potentially be maturity and sex differences. However, this notion lacks any supporting rationalisation. The authors are encouraged to either justify why there would be a maturity or sex difference that could have interacted with the findings of the current study or remove these two lines.

Conclusion

Line 405-406: Again, this is too general, a positive effect on CMJ jump height or 20m sprint time. It is not a valid statement to base the entirety of a youth’s motor skill performance on these two measures.

Line 408-409: Again, it is unclear how ADCs were more effective than ECs and ICs, and in what way. What were the measurements? What was the magnitude of the effect?

Line 414-426: These two paragraphs seem to contradict each other. The Line 414 paragraph highlights that ECs and ADCs will only work in adults or more ‘developed’ youth participants; whereas, the Line 423 paragraph highlights that ECs and ADCs should still be used with youth populations. A conclusion that fits with what has been measured is that ECs, ICs, and ADCs do not seem to positively affect CMJ jump height or 20m sprint time. However, these findings should not deter practitioners from using ECs, ADCs, or ICs with youth populations as all cueing strategies help with the learning and performance of a motor skill. Similarly, more research is needed to see how ECs, ICs, and ADCs affect the performance and learning of motor skills by including more motor skill tasks and extensive measures.

7. PLOS authors have the option to publish the peer review history of their article (what does this mean?). If published, this will include your full peer review and any attached files.

Reviewer #2: **Yes: **Saldiam R. Barillas

---

## [Author Response · Author response to Decision Letter 1]

29 Nov 2022

We would like to sincerely thank the reviewer once again for the time that they have taken in evaluating the revised manuscript and for the recommendations that were put forward. Below we have provided an itemised response to each comment. In the vast majority of cases we were able to incorporate the reviewer’s recommendations into the paper and we extend our thanks for their help in improving the quality of the article.

Regards

Jason

Reviewer #2: Review of Manuscript: PONE-D-22-21775 Do verbal coaching cues and analogies affect motor skill performance in youth

populations?

Title: Do verbal coaching cues and analogies affect motor skill performance in youth

populations?

The authors have written a manuscript investigating cues of external (EC) or internal (IC) focus, with an additional condition of analogies with a directional component (ADC) on sprint and jump performance in youth performers. A control cue condition was also included.

The authors have addressed a majority of the comments. There are still a few general comments and specific comments that would strengthen the rigour and quality of the manuscript.

Below are general comments for the author's consideration.

General Comments (GC):

Reviewer #2: GC1: The tables are not fully legible as they are cut off in the submission.

Response: This had to do with the portrait/landscape orientation of the pages which we have now remedied. The tables should now be fully viewable.

Reviewer #2: GC1: GC2: The authors are encouraged to stop using “we” and “our” in the manuscript and maintain an objective (3rd person) writing style throughout.

Response: This has been changed and edited in each instance in the manuscript with sentences amended according to the context.

Reviewer #2: GC3: The discussion does little to discuss how CMJ jump height or 20m sprint performance has previously been affected by cueing interventions. Or why no benefits were observed in these particular measures. The authors are encouraged to discuss their measures.

Response: We have now reviewed the discussion and added areas of text that align with the reviewer’s recommendation here.

Firstly, we have used the review of Barillas et al. (2021) to strengthen our choice of hypotheses in the discussion section. We feel this is a more efficient way to incorporate the type of research highlighted by the reviewer without having to cite all of the research studies in this area:

“Previous research has demonstrated that coaching language, such as ECs, can have a positive effect on sprint and jump performance; however most evidence relates to adult rather than youth populations (Barillas et al., 2021). Accordingly, it was hypothesised that ECs, ICs and ADCs would be more effective than neutral control cues for enhancing jump and sprint performance, that ECs would be more effective than ICs and that ADCs would be more effective than both ECs and ICs.”

We have also used cited the review article of Barillas et al. to reinforce the reviewer’s first point of this recommendation. We feel this is a more efficient way to summarise the literature in this case.

“However, as youths develop as they age, their neurocognitive capacities, as well as their ability and willingness to follow instructions, can vary (Kushner et al. 2015). This means that when working with youths, the challenges of coaching may be different to those encountered when working with adults (Kushner et al. 2015). The results of the current study are indicative of this when compared to current literature which indicates that coaching techniques such as ECs result in enhanced sprint and jump performance in adults and youths, though evidence in the latter group is relatively scarce (Barillas et al., 2021).”

The reasons the reviewer might have the impression that the underpinning mechanisms for results are not discussed in extensive depth is that we freely admit in the paper that we are not entirely sure why they occurred in this manner across some fairly heterogenous male youth populations. Quoting ourselves:

“Based on the extant evidence, these concepts seem to hold in adults (35) yet according to the current results at least, the predicted outcomes of constrained action hypothesis could be impeded in certain youth populations in performance tests such as the vertical jump and 20 m sprint. Why this is the case is not entirely clear; however, there could be a practical explanation for this finding that has previously been raised by Maxwell and Masters (36).”

We then go on to elaborate on the work of Maxwell and Masters citing youths’ possibly shorter attention span offsetting the effects of the cues”

These authors demonstrated that when asked to perform a balance task, performers who were provided with an internal focus of attention had switched to an external focus once they executed the task. Comparative research (37) that has been carried out in children and adults is suggestive of a shorter span of attention in the former group. The reason for this could potentially be explained by the rate of cognitive development in children and adolescents whose frontal lobes continue to mature as they grow (37).

However, in light of the relative lack of research in youths (also highlighted in the review by Barillas et al., 2021), particularly that which focuses on underpinning mechanisms, we have now also added text to the end of the discussion:

“…….research should also investigate the potential underlying mechanisms that determine attentional focus during motor skill execution in youth performers”.

Elsewhere, in the discussion we elaborate even further on our results, attempting to remain tethered to a comparison to the bulk of the research on this topic, which has been carried out in adults. We believe this keeps the discussion grounded and ensures that only valid comparisons are made. Accordingly, we state:

“Given that there were no differences in performance when ECs and ICs were used, the results of this study could point to the potentially narrower frame of reference that youths possess in comparison to adults which could, in turn, can have a detrimental impact on their understanding of a given coaching cue (12). It has been argued that because younger people have fewer past life experiences than adults, they could be classified as ‘naïve perceivers’ (40). Accordingly, young peoples’ life experiences may not have developed to the extent that they can contextualise instructions in the same way that adults do, particularly if those instructions are accompanied by an analogy that lacks context in terms of their understanding of a particular coaching cue. Moreover, whilst adults have been shown to focus on relevant cues only, children focus on both relevant and irrelevant cues and this could potentially impact on the level of attention they devote to a specific instruction (40).

The above point is further elaborated on with input from the reviewer (please see below) but we must point out that we do feel we have discussed the relevant parts our findings sufficiently and in line with the extant evidence. Moreover, we must be careful in assuming too much with our results. Elsewhere in this review, the reviewer has suggested that we may have been overly speculative – an assertion that we are in agreement with. We now feel we have put forward well-reasoned arguments as to why no benefits were observed in the measures and with the reviewer’s assistance elsewhere in the discussion (exemplified by our below responses) have improved the paper substantially. On this basis, we have tried to strike a balance between the reviewer’s suggestion here and our own interpretation of our results and the reasons for them. We hope the reviewer can appreciate our perspective on this and there are some very important changes that we do implement as detailed below. 

Specific Comments:

Introduction

Reviewer #2: Line 100: Authors are encouraged to include ‘might’ or ‘may’ between ‘therefore’ and ‘not’.

Response: Now edited. The sentence now reads:

“Accordingly, coaches therefore might not be using optimal methods when attempting to drive motor learning in these populations.”

Reviewer #2: Line 111-122: To strengthen the evidenced based practice angle of the paper, authors are also encouraged to review and include the following citations in this paragraph:

Radnor JM, Moeskops S, Morris SJ, Mathews TA, Kumar NT, Pullen BJ, Meyers RW, Pedley JS, Gould ZI, Oliver JL, and Lloyd RS. Developing Athletic Motor Skill Competencies in Youth. Strength Cond J 42: 54-70, 2020.

Kushner AM, Kiefer AW, Lesnick S, Faigenbaum AD, Kashikar-Zuck S, and Myer GD. Training the developing brain part II: cognitive considerations for youth instruction and feedback. Curr Sports Med Rep 14: 235-243, 2015.

Both articles indicate that analogies should be prioritised when working with younger or less trained athletes. Particularly due to the limited abstract processing skills that will develop throughout maturation and mostly in adolescents. This better describes the mechanism at work which is inferred by Fasold et al.

Response: Thank you for this recommendation and the associated references. We have now consulted both and included each in an amendment to this sentence which now reads as follows in line with the reviewer’s recommendation:

“Recently, Fasold et al. (2020) found that children exhibited improved performance in handball skills when coaching cues were delivered in an analogical format. Indeed, it has been suggested that this method of coaching delivery should be prioritised when working with young athletes on the basis that it can improve information processing by enhancing the recall of instructions, thus making those instructions more relatable to the task to which they specifically refer (Kushner et al., 2015; Radnor et al., 2020).”

Reviewer #2: Line 126: Authors are encouraged to either maintain ‘toward’ and ‘away’ in quotations or no quotations. Currently, ‘towards’ is in “” whereas ‘away’ is not.

Response: We have now amended this to read ‘toward’ and ‘away’ for consistency’

Reviewer #2: Line 129-132: Analogies combined with ECs are likely not a new tool, but rather the evidence to support its use is new. The authors are encouraged to change the wording to highlight the novelty of the understanding, not the novelty of the practice. As previous literature recommends using simpler language, analogies, and metaphors, but based on cognitive limitations and not quantified changes.

Response: We have now modified this sentence to reflect the reviewer’s recommendations:

“This could have important implications for coaching and learning as the combination of ECs and analogies could represent a previously known, yet untested tool, for coaches that could enhance the contextual understanding of a performer and, by extension, the learning of key movement skills.”

Methods

Reviewer #2: Line 201-202: As per the comment where the control cue was given more exposure than the other cues, the reviewer mistakenly read the original paragraph to read that participants were reminded to jump as high as they can with each jump. However, in the new iteration, it is now referenced here that “Prior to the jumps taking place, each participant was individually requested to ‘jump as high as you can in the remaining ten jumps’”. It is unclear, but would an athlete have experienced this format:

1. Jump as high as you can in the remaining ten jumps

2. EC / IC / ADC / Control

3. Jump 1

4. EC / IC / ADC / Control

5. Jump 2

6. EC / IC / ADC / Control

7. Jump 3

8. Etc…

Response: Again here, we were perhaps not clear enough in how we described the experimental protocol but to answer the reviewer’s question in short: no, the participants would not have experienced this format of instruction. In the format put forward by the reviewer, it appears that each cue is delivered for each jump. This was not the case. There were 10 cues and each was associated with just 1 jump (so 10 jumps in total, 1 cue per jump). A full jump protocol would have looked like this:

• Before any cue was delivered or any jump/sprint took place, the following was communicated to the participants about the protocol in general (i.e. not in relation to one specific jump or sprint):

o “Please jump as high as you can in the remaining ten jumps. Prior to each jump you will be given a specific coaching cue. Focus as hard as you can on this cue during the jump”

• They then would have been given a random jump scheme that may have looked like the below, let’s say it was ‘scheme 9’ (the order within which the cues were also randomised, ensuring double randomisation of both schemes and cues):

• Jump 1: “jump as if you are trying to catch a ball overhead at its highest point"

o Athlete then jumps

• Jump 2: “jump as if you are trying to catch a ball overhead at its highest point"

o Athlete then jumps (the randomisation process is what accounted for jump cues 1 and 2 being identical in ‘scheme 9’ but this pattern does not continue as seen below).

• Jump 3: “jump as if the ground is suddenly hot and you have to get off it as quick as possible"

o Athlete then jumps

• Jump 4: “as you jump, focus on extending your legs"

o Athlete then jumps

• Jump 5: "jump as high as you can" (CONTROL CUE)

o Athlete then jumps

• Jump 6: “as you jump, focus on extending your legs”

o Athlete then jumps

• Jump 7: “as you jump, focus on pushing the ground away”

o Athlete then jumps

• Jump 8: "jump as high as you can" (CONTROL CUE)

o Athlete then jumps

• Jump 9: “as you jump, focus on pushing the ground away”

o Athlete then jumps

• Jump 10: “jump as if the ground is suddenly hot and you have to get off it as quick as possible”

o Athlete then jumps

On the basis of the above confusion, we have now amended the wording to state “Prior to any jumps taking place”………as below:

“Prior to any jumps taking place, each participant was individually requested to “jump as high as you can in the remaining ten jumps”

We have also done this for the 20-m sprints.

Reviewer #2: If so, the neutral cue of 'Jump as high as you can' is still given a net 1 more times than the previous (this is similar to the delivery of cues for the Sprint task too). 

Response: We can see what the reviewer saying here, however, we do feel that the effect of this is almost entirely negligible, a point we hope is supported by the elongated format we have shown above which outlines a full example order in which the cues were given. For example, because of our double randomisation process, the delivery of the first general instruction can theoretically be 9 or 10 specific cues removed from the delivery of the actual control cue. The participants in the study were very well aware of which words were simply preparatory in nature and which words constituted the actual control cue to perform in response to. All were well briefed prior to the study being undertaken and the protocol was carried out with precision at each of our centres.

Reviewer #2: Similarly, it is delivered in a more natural format. To be a control for the IC and EC, it should have been delivered as “As you jump, focus on getting as high as you can” rather than “Jump as high as you can”. Or alternatively, the IC be, “Extend your legs (powerfully/forcefully/quickly)” and the EC be, “Push the ground away (powerfully/forcefully/quickly)”. This mismatched control cue should be highlighted as a limitation. However, the reviewer respects that this area of research is still being defined and parameters are still being understood. Interestingly, the results point to the control cue providing the most benefit, and it’s the cue that would be most commonly used in practice and reduces cognitive processing strain with less irrelevant information.

Response: Yes, this is a good point and we can see what the reviewer is saying here. Despite this, we don’t think it would have affected the results to any great extent (particularly given that most cues seemed to yield similar performances regardless of cue, task, place or language with a few exceptions). In agreement with the reviewer, we have now highlighted this as a limitation and, indeed, will take it on board for future work we do in this area. The limitation, alongside the pre-existing text now reads:

“Similarly, the ECs and ICs required the participants to retain a specific focus for performance whereas the neutral cues simply requested maximal performance. This small differential could impact on an individual’s understanding of a particular cue and though it was deliberate in nature, researchers must work to standardise cues across various tasks to ensure the most effective form of communication.”

Just to give a little insight on the issue, we deliberately based the control cue on previous research in this area (25,26) and we felt the cue itself needed to be as specific to the task possible. For example, “sprinting as fast as one can” is an entirely clear and possible task to achieve – the participant merely performs as best they can. Similarly, focusing on “driving the ground back” is entirely possible but literally doing it is a different matter altogether! We admit that we are delving into semantics on this point but we feel there are multiple ways in which words and language can either differ or be similar and it is a difficult balance for coaches and researchers alike.

This is reinforced by our pre-existing recommendation for more work in this area:

“Accordingly, alternative cues with different compositions, and in other languages, should be tested to examine the most effective coaching terms to enhance performance in young individuals.”

We hope the above serves as a rationale as to how we approached this work.

Results

Reviewer #2: The authors have kept the succinctness of the results with the additional text. It illustrates the absence of an effect across intervention cues.

Response: Thank you again for your acknowledgement of this.

Discussion

Reviewer #2: Line 319-321: Authors are encouraged to include Kushner et al (2015) to support this statement.

Response: We have now added the Kushner reference to this sentence.

Reviewer #2: Line 339-341 & Line 348 -351: The authors make a general statement that their results showcase the constrained action hypothesis may be impeded by youth populations. However, this is a very general statement and is only true for naïve youth populations with measures of akimbo CMJ jump height and 20m sprint time. The authors are encouraged to be more specific as research currently exists that young (~11-year-olds) and old (~16-year-olds) youth athletes benefit from the constrained action hypothesis (see references):

Barillas, S. R., Oliver, J. L., Lloyd, R. S., & Pedley, J. S. (2022). Kinetic Responses to External Cues Are Specific to Both the Type of Cue and Type of Exercise in Adolescent Athletes. Journal of strength and conditioning research. (published ahead of print)

Oliver, J. L., Barillas, S. R., Lloyd, R. S., Moore, I., & Pedley, J. (2021). External Cueing Influences Drop Jump Performance in Trained Young Soccer Players. Journal of strength and conditioning research, 35(6), 1700–1706.

Response: In relation to lines 339-341, we have now been more specific with the language to reflect the recommendations of the reviewer. The lines are now rewritten as such:

“Based on the extant evidence, these concepts seem to hold in adults (37) yet according to the current results at least, the predicted outcomes of constrained action hypothesis could be impeded in certain youth populations in performance tests such as the vertical jump and 20 m sprint.”

In relation to lines 348-351, we have also amended the wording to state that this may apply to certain naïve youth groups and we have now cited Barillas et al. (2022) and Oliver et al. (2021) as evidence that counteracts that which we present in the current study. We hope this gives a more rounded view of the literature as it currently stands and acknowledges the important work of others in this growing domain of research:

“Whilst concepts such as the constrained action hypothesis might serve as an effective model for motor performance and learning in adults, an alternative approach could be more appropriate in certain (perhaps naïve) youth groups due to the aforementioned factors, though evidence to the contrary does exist (Barillas et al., 2022; Oliver et al., 2021).”

Reviewer #2: Line 355-368: The paragraph seems to continue the argument that these young participants were unable to realise the benefits of the constrained action hypothesis. However, the argument seems to put the weighting on the participants, as they were unable to focus on only the relevant information. Rather, it could be argued that they were unable to realise the benefits of the constrained action hypothesis because they were given irrelevant information, thus blunting a potentially beneficial effect. This is supported by the neutral cue receiving the most benefit in CMJ jump height. The paragraph seems appropriate to discuss this limitation, rather it implies that only adults can benefit from external cues. Again, see the previous two references. It could be argued that if they were more trained and had more exposure to the cue to no longer be ‘naïve’ then maybe they would receive a benefit. However, in the current state, the paragraph does not read this way.

Response: This is a very good point by the reviewer and very much understand the angle they come from. Accordingly, we have now added text (and the Barillas et al. 2022 reference) to this paragraph in line with the recommendation of the reviewer which reads as such:

“On the contrary, it is important to consider that if a young individual achieves a sufficient volume and quality of training, it is possible that they would no longer be considered to be a naïve perceiver and so may respond more readily to ADCs or ECs when provided by a coach (Barillas et al., 2022)”

In addition to this, we have also deleted some of the text relating to adults so as to soften our perhaps overly-stated point here. The following sentence has now been taken out of the manuscript:

“On this basis, if presented with an ADC cue that is intended to elicit a specific motor response, that response might be more likely to be observed in an adult, who might have a better contextual understanding of the scenario that is verbally described by the coach.”

Reviewer #2: Line 370-386: This paragraph is discussing how ADCs might be an effective way of driving motor performance. However, the only two measures included in the study were CMJ jump height and 20m sprint time. What measures are included in the study to come to the conclusion that ADCs might improve motor performance? If there are measures of motor performance or perception of the cues, please include them in the methodology and results. If not, this paragraph needs to be removed as it would then be entirely speculative.

Response: We can see what the reviewer is saying here and the problem may have been caused by our somewhat inaccurate (perhaps even verbose?!) way of expressing ourselves in this paragraph. First off to address the reviewer’s concerns, we have now amended the terms here, replacing ‘motor performance’ with the specific names of the tests that were actually conducted, those being the CMJ and the 20 m sprint. As an example, we now state instead:

“We did observe limited evidence that ADCs might be an effective way of driving improved performance in sprinting and jumping in young individuals.”

The above is a factually true statement that is reflected by our results and so we believe it is worth discussing in the manuscript. Also, in line with the reviewer’s recommendation, we have now deleted part of the paragraph in question. We are in agreement that the below sentences are too speculative in nature and are probably not within the scope of the study. Upon re-reading, we do agree that this text is misplaced:

“The practice of storytelling to increase both sport and exercise participation in youths has previously been utilised with characters from Disney movies being used as a prism through which children can contextualise their experiences in physical activity (19–21). In a similar vein, it appears that ADCs can be used to achieve a greater level of engagement in youths.”

Coming back to our above observation that our results are factual and thus must be discussed in the discussion section, we do not necessarily agree 100% that the paragraph is too speculative and that with the above amendments it is now more in line with the current literature in this domain. Indeed, even if it were speculative, we do believe that there is room for such reasoned speculation in a discussion section so as to offer potential explanations for the observed results and potential future avenues of investigation. This happens all the time in strength and conditioning based studies which include a marker of performance, but perhaps no marker of potential mechanism(s).

The new paragraph, pasted below, does not contain a single statement that is not supported either by our own current results or the results and/or opinions of other prominent authors in this field. The newly edited paragraph does not use language or terms that are unequivocal in nature, meaning we are open to questioning on each claim that we make (as we are through this current peer-review process). We feel we have avoided any hyperbolic claims and have adhered to common practice and evidence-based results throughout the piece. Perhaps more importantly however, what remains of the previous paragraph can be of use to coaches. As coaches ourselves, we wish to engage with research that not only speaks to the academic community, but also the coaching community. In this paper we seek to do this and so the promotion of sound coaching practice is central to the discussion of our results. We do hope the reviewer can appreciate our stance on this and the new paragraph now reads as such:

“We did observe limited evidence that ADCs might be an effective way of driving improved performance in sprinting and jumping in young individuals. The use of analogies in coaching youths may serve as a more relatable model of communication that facilitates a better understanding of a coach’s cue than the use of traditional biomechanical terminology (35). In this way, the evocative language of instructing a young athlete to “take off like a rocket” (6) could be preferable if it results in a better contextual understanding than a cue relating to the movement and angles of specific limbs. However, where ADCs could potentially have drawbacks is in relation to the kinetic and kinematic characteristics of a given movement. For example, Winkelman states that if the information verbalised by a coach is not related to the task-relevant characteristics of a given motor skill, a cue may be less effective (11). Moreover, as an analogy can contain several different pieces of information in a single cue, it could have a negative effect on working memory during performance (41). Accordingly, the relevance of the information provided, relative to the action being performed, appears to be vital in driving sprint and jump performance in young individuals.”

Reviewer #2: Line 388-403: The limitation paragraph briefly discusses that the lack of standardization within neutral/control cues could have led to the observed benefit. Rather, the current evidence in the field points to the neutral control cue condition having less irrelevant information, delivered in a manner more akin to a way a coach would do in practice, and potentially highlighting an extra time at the beginning of each set of jumps or sprints. It is really hard to argue that these youth populations, all of whom play sports (soccer or rugby) aside from two school groups, would not be naïve to a 20m sprint. This paragraph should take the lack of standardisation argument towards future research using content-matched cues.

Response: We have already added to this limitations paragraph based on other recommendations of the reviewer but we have now further strengthened it based on the comments put forward by them here. Accordingly, we have now acknowledged the importance in considering the combination of the cue type and the experience level of the performer:

“An important consideration here is for researchers to compare the effects of content-matched cues in both naïve and non-naïve populations alike as the results could be different in each based on different combinations of the type of cue delivered and the level of experience of the performer”.

Reviewer #2: Line 400-403: These lines of future research indicate that there may potentially be maturity and sex differences. However, this notion lacks any supporting rationalisation. The authors are encouraged to either justify why there would be a maturity or sex difference that could have interacted with the findings of the current study or remove these two lines.

Response: We have now deleted these lines in line with the reviewer’s recommendation.

Conclusion

Reviewer #2: Line 405-406: Again, this is too general, a positive effect on CMJ jump height or 20m sprint time. It is not a valid statement to base the entirety of a youth’s motor skill performance on these two measures.

Response: In line with this comment we have now amended this sentence to reflect the reviewer’s recommendation: 

“Our results did not generally indicate that the type of experimental cue used had a positive effect on performance tests such as the vertical jump and 20 m sprint in youths.”

Reviewer #2: Line 408-409: Again, it is unclear how ADCs were more effective than ECs and ICs, and in what way. What were the measurements? What was the magnitude of the effect?

Response: We appreciate this and have now added the detail of what the measurements were. The magnitude of the effect is already reported elsewhere in the paper so is not repeated here.

We have now edited the text to read:

“There was no evidence to support the initial hypotheses that ECs, ICs and ADCs would be more effective at enhancing vertical jump and 20 m sprint performance than neutral control cues, or that ECs would be more effective than ICs for the same measures.”

And just below that, we have added further detail:

“There was however some limited evidence that ADCs are more effective than both ECs and ICs at enhancing 20 m sprint performance.”

Reviewer #2: Line 414-426: These two paragraphs seem to contradict each other. The Line 414 paragraph highlights that ECs and ADCs will only work in adults or more ‘developed’ youth participants; whereas, the Line 423 paragraph highlights that ECs and ADCs should still be used with youth populations. A conclusion that fits with what has been measured is that ECs, ICs, and ADCs do not seem to positively affect CMJ jump height or 20m sprint time. However, these findings should not deter practitioners from using ECs, ADCs, or ICs with youth populations as all cueing strategies help with the learning and performance of a motor skill. 

Similarly, more research is needed to see how ECs, ICs, and ADCs affect the performance and learning of motor skills by including more motor skill tasks and extensive measures.

Response: Thank you for this recommendation. We have incorporated virtually all of it into our conclusion, though have not pasted it here as it is a little lengthy for response document.

---

## [Decision Letter · Decision Letter 2]

22 Dec 2022

Do verbal coaching cues and analogies affect motor skill performance in youth populations?

PONE-D-22-21775R2

Dear Dr. Moran,

We’re pleased to inform you that your manuscript has been judged scientifically suitable for publication and will be formally accepted for publication once it meets all outstanding technical requirements.

Kind regards,

Filipe Manuel Clemente, PhD

Academic Editor

PLOS ONE

Additional Editor Comments (optional):

Reviewers' comments:

Reviewer's Responses to Questions

**Comments to the Author**

1. If the authors have adequately addressed your comments raised in a previous round of review and you feel that this manuscript is now acceptable for publication, you may indicate that here to bypass the “Comments to the Author” section, enter your conflict of interest statement in the “Confidential to Editor” section, and submit your "Accept" recommendation.

Reviewer #2: All comments have been addressed

2. Is the manuscript technically sound, and do the data support the conclusions?

Reviewer #2: Yes

3. Has the statistical analysis been performed appropriately and rigorously? 

Reviewer #2: Yes

4. Have the authors made all data underlying the findings in their manuscript fully available?

Reviewer #2: Yes

5. Is the manuscript presented in an intelligible fashion and written in standard English?

Reviewer #2: Yes

6. Review Comments to the Author

Reviewer #2: Review of Manuscript: PONE-D-22-21775 Do verbal coaching cues and analogies affect motor skill performance in youth

populations?

Title: Do verbal coaching cues and analogies affect motor skill performance in youth

populations?

The authors have written a manuscript investigating cues of external (EC) or internal (IC) focus, with an additional condition of analogies with a directional component (ADC) on sprint and jump performance in youth performers. A control cue condition was also included.

The authors have addressed all the comments with excellent attention to detail and argument points I would agree, strengthen your rationale, but also clarify any confusion from the reviewer.

Thank you for taking the time to clearly respond to each comment, this is a great addition to the cueing body of research in a strength and conditioning setting.

7. PLOS authors have the option to publish the peer review history of their article (what does this mean?). If published, this will include your full peer review and any attached files.

Reviewer #2: **Yes: **Saldiam R Barillas

---

## [Editor Report · Acceptance letter]

22 Feb 2023

PONE-D-22-21775R2 

Do verbal coaching cues and analogies affect motor skill performance in youth populations? 

Dear Dr. Moran:

I'm pleased to inform you that your manuscript has been deemed suitable for publication in PLOS ONE. Congratulations! Your manuscript is now with our production department. 

Kind regards, 

on behalf of

Dr. Filipe Manuel Clemente 

Academic Editor

PLOS ONE